# Evaluation of Geometric Data Registration of Small Objects from Non-Invasive Techniques: Applicability to the HBIM Field

**DOI:** 10.3390/s23031730

**Published:** 2023-02-03

**Authors:** Juan Moyano, Elena Cabrera-Revuelta, Juan E. Nieto-Julián, María Fernández-Alconchel, Pedro Fernández-Valderrama

**Affiliations:** 1Department of Graphical Expression and Building Engineering, University of Seville, Ave. Reina Mercedes, 4A, 41012 Seville, Spain; 2Department of Mechanical Engineering and Industrial Design, University of Cádiz, 10 Avenue Puerto Real, 11510 Cádiz, Spain

**Keywords:** BIM small object, structured-light systems, 3D optical scanner, point cloud, sculpture, cultural heritage

## Abstract

Reverse engineering and the creation of digital twins are advantageous for documenting, cataloging, and maintenance control tracking in the cultural heritage field. Digital copies of the objects into Building Information Models (BIM) add cultural interest to every artistic work. Low-cost 3D sensors, particularly structured-light scanners, have evolved towards multiple uses in the entertainment market but also as data acquisition and processing techniques for research purposes. Nowadays, with the development of structured-light data capture technologies, the geometry of objects can be recorded in high-resolution 3D datasets at a very low cost. On this basis, this research addresses a small artifact with geometric singularities that is representative of small museum objects. For this, the precision of two structured-light scanners is compared with that of the photogrammetric technique based on short-range image capture: a high-cost Artec Spider 3D scanner, and the low-cost Revopoint POP 3D scanner. Data capture accuracy is evaluated through a mathematical algorithm and point set segmentation to verify the spatial resolution. In addition, the precision of the 3D model is studied through a vector analysis in a BIM environment, an unprecedented analysis until now. The work evaluates the accuracy of the devices through algorithms and the study of point density at the submillimeter scale. Although the results of the 3D geometry may vary in a morphometric analysis depending on the device records, the results demonstrate similar accuracies in that submillimeter range. Photogrammetry achieved an accuracy of 0.70 mm versus the Artec Spider and 0.57 mm against the Revopoint POP 3D scanner.

## 1. Introduction

There are several strategic lines in Cultural Heritage (CH) dealing with the preservation, conservation, restoration, and maintenance of the built environment. One of them is research, as it contributes to the knowledge of movable and immovable assets. The development of techniques such as digital photogrammetry and the availability of structured-light scanning equipment play an essential role in the capture of objects and works of art. 3D scanning applied to archeological objects or artifacts allows the shapes and their texture to be stored in a digital format, speeding up the process of representation, identification, and cataloging. On the other hand, with the new disruptive building information modeling technologies applied to archeology and historical architecture, the creation of parametric 3D models of those artifacts poses a challenge for these areas of knowledge. At the same time that these new technologies appeared, the complexity of these processes in the conservation of cultural assets has required new studies and applications to achieve UNESCO’s recommendations; these are the protection of landscapes, natural environments and those created by humankind, which have a cultural or aesthetic interest, or which form a harmonious natural whole [1]. Choosing the right technology and equipment allows operators who work with digital tools to follow an effective workflow and achieve high-quality results. Nowadays, low-cost 3D sensors are the most popular for entertainment purposes, but they can also be used in research. Identifying the possibilities of this technology for small objects is a current challenge in society and research in the fields of archeology and architecture.

State-of-the-art geodetic measurement methods, whether through passive sensors or active sensors, are used to obtain accurate and quality 3D records. There are few studies in which the Terrestrial Laser Scanner (TLS) is used for small objects since it is mainly used to record art works. State-of-the-art passive sensors, such as image capture techniques or active structured-light sensors, are used in most of the studies into the precision and quality of the recording of small objects. Thus, a good part of the engineering sector uses the technology of re-engineering and control of parts of objects as in the field of aeronautics for technological manufacturing equipment [2]. Research has also compared the growth and erosion results of tufa obtained using hand-held 3D scanners [3], and has addressed the evaluation of a small sculpture by the Gallery of Matica Srpska to protect 3D models of the original sculpture [4]. Therefore, long-range active sensors are less used in sculpture, works of art, or other small objects. The use of the Structured-Light Scanner (SLS) technique is more common, but can be an example of long-range active sensors [5]. The latter technology provides submillimeter-scale resolution, but has a limited field of view, while TLS provides millimeter-scale resolution, but requires equipment that is expensive for most professionals or academics. In addition, TLS requires experts to control not only the capture, but also the post-processing of the records. In this sense, there is also a work effort to align the TLS point clouds [6].

The improvement of the image capture algorithms allow progress to be made over time, although there are gaps where work is currently being conducted to improve the robustness, accuracy [7], integrity [8] and scalability [9] of the final model. The use of small object studies for reverse engineering in industry [10] and in the medical sector [11,12,13,14], initiates the analysis of sculptures and small objects in the CH field. When low-cost scanners appeared, researchers tried to identify the process and methodology for the acquisition of geometric shapes [15]. The transportability of low-cost structured-light scanners is their main advantage when working in museums and archeological sites. An example of their use in archeology is the work by McPherron et al. [16], who achieved submillimeter precision by controlling the lighting. In these image sensor technologies, the integration of optical and electronic components plays an essential role [17]. Thus, in the field of studying architectural heritage, the integration of information from multiple sensors is also studied, combining metric data with temperatures and the creation of a 3D thermal texture [18,19]. Or the analysis of the texture of the color and 3D shape [20], and the changes in the deformations of the Badillo et al. [21] paintings. Most of the studies dealing with small archeological objects focus on 3D implementation. Li and Zha’s work [22] on 3D virtual restoration of archeological and cultural heritage used this technology to create digital files and 3D lines [18]. Their unpublished results could be useful for future intervention projects. The creation of 3D lines unveils geometric or organic figures that determine impossible drawings in the sculptures. Three-dimensional virtual reconstruction makes it possible to obtain orthophotos and the Digital Elevation Model (DEM) or Digital Terrain Model (DTM) from the 3D model that integrates detailed and precise information in the digital plan of the archeological excavation [23]. Other studies analyze the 3D model of archeological objects [23,24,25,26] to reveal their geometrical similarity. Focusing on the comparison between massive data acquisition techniques, Molero et al. [27] evaluated the use of structured-light scanning and photogrammetry through an Artec MHT 3D scanner. Kersten et al. [28] compared the geometric accuracy of portable 3D scanning systems, including the Artec Spider, with other equipment with similar features and prices. Other works dealt with RGB-D cameras such as the Kinect v1 (Microsoft), which was released on the market around 2014. Lachat et al. [29] evaluated the accuracy of this sensor for 3D reconstruction of small objects. The Artec Spider and the Revopoint POP 3D scanner have been used [30] to evaluate the size change of an implant in two skulls for additional surgical purposes, a plastic model and a human model. Morena et al. [31] worked on the precision of the low-cost EinScan-Pro in a sculpture by Eduardo Chillida. Three-dimensional modeling of small archeological objects requires an effective methodology to capture minor geometric details [1]. Another challenge is to capture the bottom of cylindrical objects. The numerous new scanners, as well as camera features and their calibration are also noteworthy [5]. As far as it has been investigated, no work has evaluated these data acquisition technologies, 3D models developed in a BIM environment, nor the precision of low-cost 3D sensors for their suitability for the virtual reconstruction of objects in museums and academic spaces. Therefore, the work provides information on the precision of the equipment, through the evaluation of algorithms and through the study of the density of points at a submillimeter scale. Thus, this paper aims to evaluate and compare the recording precision of two structured-light scanners against Structure from Motion (SfM): the high-cost Artec Spider 3D scanner and the low-cost Revopoint POP 3D scanner. To do this, the focus is on a small archeological artifact with geometrical singularities that is representative of small museum objects and pieces. The applicability of these fast and accurate data capture technologies is useful for creating predictive degradation models, evaluating sculpture painting, and detect pathologies related to stone mineralogy. Furthermore, a digital twin of the object developed in a Building Information Modeling (hereinafter, BIM) environment is evaluated in this research. Point cloud data from 3D scanning and photogrammetry are used to create parametric BIM objects of complex shapes in the artistic sculpture domain. This evaluation process is carried out in two different environments, one under a comparison algorithm used in scientific research such as Cloud Compare and a new framework through the BIM methodology; this workflow is described in the Figure 1.

## 2. Case Study

The methodological focus on the study of the different technologies that can contribute to records of archeological objects contemplates the capture of data to a small vessel of 28 cm in diameter by 20 cm high. In the development of the vessel in its outer part, the figures of the twelve apostles are represented. The work is made of low-density sandstone and the piece is uncatalogued. The integration of several 3D modeling technologies makes the work unprecedented, so a workflow is configured using photogrammetry techniques performed with GCP through control points. The records of the 3D optical measurement systems were carried out through the respective software that is supplied by the equipment manufacturers.

The study focuses on four types of comparisons: (i) compare the degree of metric-dimensional precision through an Iterative Closest Point (ICP) algorithm, defined by Besl and Mckay [32], (ii) analyze the density of points of the subsets obtained from the geometric analysis, (iii) evaluate the quality of the mesh of each of the models in micro surfaces, (iv) analyze the precision between models through vector theory in the integration of points in BIM, and finally (v) establish the best procedure to bring an archeological-architectural object to BIM. In order to validate the accuracy of section (i), a study of a second case study of a plaster sculpture was carried out, which was made by students of the Faculty of Fine Arts at the University of Seville.

## 3. Sensor Characteristics and Data Acquisition

This section provides the available data from the manufacturers and the procedures of the 3D optical measurement and photogrammetry image capture systems. Both techniques, structured light scanners, and photogrammetry, acquire information about the geometry of objects with complex shapes and capture information about the color generating the texture of the objects. From the data collection of the Artec Spider scanner, a cloud of points called SLS_VS_ is obtained, in the same way, from the data of the Revopoint POP 3D scanner, a set of points called SLS_VP_ is obtained and finally through the technique of capturing images through photogrammetry the set of SfM_V_ points is obtained.

### 3.1. Artec Spider 

The Artec Spider scanner is a device designed to obtain models of scientific objects in three dimensions using structured light technology (speckle pattern) with blue LEDs as the light source. It features 3D resolution down to 0.1 mm, 3D point accuracy down to 0.05 mm, and 1.3MP texture resolution. This equipment requires a working distance between 0.2 and 0.3 m and provides a scanning area of 90 × 70 mm (height × width in the closest range) and 180 × 140 mm (height × width in the farthest range). The measuring range sensors are between 0.17 and 0.35 m, and data acquisition speed, up to 1 min points/seg. The 3D shape measurement system setup and system calibrations has been verified by Zhao et al. [33]. The scanner has been designed for registration of complex surfaces, sharp edges and structures with slight fluctuations. The use of a rotating platform that allows complete registration of the piece is convenient. The work has been developed in the photo library of the Art Laboratory of the University of Seville, where the Artec Studio Professional v.12 software is installed Figure 2. The registration begins by activating the software so that the scanner reaches a temperature of 50 °C for optimal reading in real time. The processing phases are developed in the same software following the following workflow: (i) registration, (ii) fusion, and (iii) postprocessing. The great advantage of the scanner is that it obtains automatic alignment and has point cloud filtering and mesh closure algorithms. This fact results in a better-quality post-processing than that obtained by low-cost scanners. This equipment has been used in different investigations, for example, for digital models of slope micro geomorphology [34] or in the heritage works mentioned above [31].

### 3.2. Revopoint POP 3D Scanner

The POP handheld scanner is a device made up of a camera and an integrated chip for fast and accurate scanning. This scanner is portable and compact, capable of scanning of various types. The Revopoint POP 3D Scanner is provided with binocular structured light, ensuring that the acquired 3D point cloud data features high accuracy. The highest single frame accuracy can reach 0.3 mm. POP meets 0.05 mm professional-grade precision and a 3D point cloud data of 0.15 mm. A set of depth cameras, with two IR sensors and one projector, can quickly obtain the 3D shape of objects, and one RGB camera is used to capture texture information. This device allows directly generating 3D models. It is suitable for both indoor and outdoor use. This scanner can be connected to a computer or mobile. The software used is Handy Scan, which is easy to use and intuitive. The scanner connects via USB to the computer and, by means of a data transfer cable, the model is exported. The file formats of “obj., stl. and ply.” are supported. 

Scanning workflow: the object is placed on the table; POP is connected to the PC and the HandySCANrogram is opened Figure 3. The object is moved at an “excellent” distance. This is performed by adjusting the parameters (auto first, then manual to obtain a proper image quality). Once finished, the 3D model is exported in all three file formats (obj., stl. and ply.).

### 3.3. Structure from Motion Survey

The structure from motion image-based data capture technique is an algorithm well-known by the scientific and academic community [35]. In archeology, it is used on numerous occasions to generate records of information in three dimensions. This work is focused on medium size pieces, that can be moved and placed on a swivel base. 

In this work, the camera used was a Panasonic DMC-GF3 with a fixed focal length of 14 mm. This camera is easily portable and compact, but also provides the possibility of changing the lens, being an intermediate solution between compact and reflex cameras. This camera has a Live-MOS sensor (17.3 × 13 mm), with 12.1 megapixels (4000 by 3000 pixels). The pixel size for this camera is 4325 µm. It is considered that the specifications of this camera are adequate for carrying out this photogrammetry work.

Due to the dimensions of the object, the data collection was carried out by placing it on a swivel base. In this kind of data acquisition process, it is advisable to fix the camera position using a tripod. The use of tripod in data collection allows two goals: the first of them is to keep the distance between object and camera, guaranteeing the acquisition by perfect circles around the object; and the second, the possibility of increasing the time of exposure of the capture, allowing to catch a high amount of light, obtaining good depths of field for sharped photos. This kind of data acquisition is well controlled, and it is possible to calculate parameters such as the distance to the object and the light in a better way. Another advantage of this kind of data acquisition process is the lack of need for it to touch the object. In many cases, objects to be reconstructed by photogrammetry must be protected. The use of swivel bases allows one to catch the geometry from different points of view without changing the position of the object and, therefore, without touching it. For long exposures, it is advisable to use remote control for capturing the image, to avoid blurred effects.

In this kind of data acquisition process, per Figure 4, in which the camera is fixed and the object is the element that changes its position, it is important to have into account that the background of the scene must be as homogeneous as possible. The fact that the object is moving and the background is fixed can generate problems in the software, which tries to match homologues points in different photos to align them. To avoid this, the best option is to prepare the scene with homogenous background and homogenous filtered light, to avoid the generation of shadows on the surface. Moreover, the software for the alignment of the photographs allow us to generate masks in those areas in which it is not advisable to look for points to match, being usual to apply mask to the background of the scene. 

To carry out a complete data acquisition, the tripod has been placed in three different positions: the first from a frontal position, the second to catch the top of the object and part of the interior, and the third to catch the interior part of the vase. The object has not been moved, for that reason the base has not been registered. 

The capture was completed with 142 images. The main specifications of the capture data are shown in Table 1.

The Ground Sample Distance (GSD) represents the resolution and details of the final 3D reconstruction [36]. The pictures were taken from a distance equal to or smaller than 0.50 m. This distance produces a GSD of 0.15 mm. 

Furthermore, on the swivel base, eight different coded markers have been placed. These coded markers are useful for three reasons. On the one hand, these markers help the alignment process, providing homologue points. On the other hand, they allow for scaling the model. To conclude, these markers facilitate the control of the quality of the result, working as Ground Control Points (GCP).

These 142 images have been processed using Agisoft Metashape, obtaining a dense cloud with 2 million points. This set is called SfMv points. The software processes the images through mathematical algorithms on 3D shapes known as Structure from Motion (SfM) [37]. As previously mentioned, the coded markers work as GCP. Therefore, the scale bars inserted between these coded markers provides a minimal error of 0.0028 mm. To obtain it, two different steps were executed.

The first step “Align cameras” was executed in the higher accuracy. As the images were taken in an ordered manner, the generic/reference preselection was activated. The parameters “Key point limit” and “Tie point limit” are set in their default values. The “Key point limit” sets the maximum number of feature points considered by the software. The “Tie point limit” is the maximum number of points that the software will match between photos. Both parameters were fixed in their default value, which are 40,000 and 4000, respectively. The parameter “Adaptive camera model fitting” is advisable to be activated as it ensures a better alignment of the cameras. This option enables automatic selection of camera parameters to be included into adjustment based on their reliability estimates. After this first step, not only the orientation of the cameras was solved, but also a sparse cloud of points was generated. Then, the “Build dense cloud” step was executed. In this step the software calculates depth maps for every image. The parameter “Quality” specifies the desired quality of the depth maps generation, being set at "High". Due to some factors, e.g., noisy or blurred images, the software offers several built-in filtering algorithms. The “Filtered mode” moderate is an intermediate algorithm of filtering. To conclude, the option to calculate point colors was activated. Table 2 sums up the configuration parameters used in this study for these two steps.

## 4. Experimental Study and Data Analysis

### 4.1. Point-Cloud Accuracy Using ICP Algorithm 

In studying the accuracy of point cloud capture, it is necessary to achieve quantitative control of coincident point measurements. This quantitative control between the results of the set of points is carried out using the open source CloudCompare software, specifically, using the Cloud-to-Cloud (C2C) tool [38]. This tool is based on the Iterative Closest Point (ICP) algorithm. Once the different clouds have been imported into this software, a first cleaning of atypical or residual points is performed manually. Subsequently, the clouds are roughly aligned, to subsequently perform an adjusted registration. For this, the closest-point algorithm is used, which recalculates the transformation parameters for the distance of the homologous points between the three sets of SLS_VS_ points; SLS_VP_ and SfM_V_.

The cleaning of residual points is carried out on the set of points of the SfM, since the registration of the two structured light models has not generated residual points. For the alignment of the models, six pairs of points have been taken (R0-A0, R1-A1, R2-A2, R3-A3, R4-A4 and R5-A5). These points were strategically located on the rim surface of the vessel. The average RMS error of the raw alignment was RMS 3.7918 mm. After registration, the average RMS error is 1.8525 mm. According to Jafari [39] a rigid transformation occurs between *pi* Ꞓ P and *qi* Ꞓ Q where *pi* Ꞓ P is a point from the 3D reference point cloud and *qi* Ꞓ Q is a point from the target point cloud. Calculate the rotation *R* and translation *t* between the two indicated points according to Equation (1).
(1)$$E\left(R,t\right)=\underset{R,t}{\min}\sum_{i}\left|\left|p_{i}-\left(R_{qi}+t\right)\right|\right|{\;}^{2}$$

The geometric deviation between the three records was calculated through the C2C comparison in Table 3, Table 4 and Table 5. In comparison procedures of this type, it is likely that there are outliers that are not visible at first sight and that can affect the results. For this reason, an iterative point filtering procedure established in evaluations is carried out, indicating the criteria for stopping the process. 

The next step determines the evaluation of two yield results of the comparison between point clouds (Figure 5), and Figure 6 that shows the histogram in which the units are expressed in millimeters and the values reach units of micrometers (1 × 10^−6^). The SLS_VS_ point set has a total of 462.020 points. 

Comparison algorithms have been used, especially in the field of engineering, to detect levels of changes in landslides that affect buildings [40], in inspection of bridge and tunnel structures [39], or historic buildings [41]. It has also been applied to the study of geomorphic changes that affect sea cliffs [42], in flood risk management [43], as well as earthquake induced landslides [44]. In short, this algorithm is applied to aspects of significant changes that can be recorded through the point cloud. However, particularly significant are the works that compare techniques [27] and profile of buildings [45] or comparison of objects [31].

The difference between Artec Spider and Revopoint POP 3D resulted in two evaluations, as shown in Table 3. The stopping criterion is established when the maximum distance in the next evaluation does not modify the previous value. It has been considered to express the different registers in the iterative process of point filtering, since in view of the scientific literature up to now we do not know of any research that has shown consecutive results when these can alter the said results. To the best of our knowledge, investigations using ICP on sculptural objects [46] or on architectural elements [7,47], as well as other comparison software such as Geomagic Studio [48] express the results as a definitive evaluation. 

In the next phase of work, the comparison between Artec Spider and the photogrammetry technique is carried out Figure 7 and Figure 8. In this, it had three evaluations, until reaching the cleaning of residual points where the results do not present significant changes as results, as shown in Table 4. 

The last phase consists of the comparison between the photogrammetry technique and the Revopoint POP scanner. The SfM_V_ point pool has a total of 1.998.068 points, and the SLS_VP_ point pool has a total of 6.403.161 points. A best fit is made with an automatic alignment through six homologous pairs, giving an RMS result of 2.32186 mm, and then the ICP algorithm is applied to optimize the alignment. Some results of three evaluations are obtained in Table 5 by means of automatic alignments with 6 points. Next, a manual pre-alignment is performed by approximating the sets of SfMV and SLSVP points with the best visual fit through cross sections and tabulating “shift box” on the respective axes (x,y,z) made in the same Cloud Compare software, this allows check how the data set works if we chose to do it manually. The results are shown in three best-fitting manual evaluations in Table 5. The contribution of this section tries to evaluate two data recording procedures <<manual and automatic>> in the Cloud Compare software and determine their action capabilities. Figure 9 and Figure 10 show the results of evaluation 3 in the automatic alignment phase. Figure 11 and Figure 12 show the results of evaluation 3 in the manual best fitting phase.

The distance between subsets of points was calculated by comparing the results of SLS_VS_, SLS_VP_ and SfM_V_. The parameters studied are the root mean square (RMS) error, the minimum and maximum distances between the point clouds, the average distance, the standard deviation and the maximum estimation error. According to Anton et al. [49], the deviation between similar objects presents two main characteristics: the high presence of points in the zero value relative to the rest of the distance intervals and the high standard deviation according to the formulation expressed by Arias et al. [50] of the points along those intervals (Equation (2)).
(2)$$\Sigma =\sqrt{\frac{1}{n-1}\sum_{i=1}^{n}(x_{i}}-\bar{x}{)}^{2}$$
where n is the sample size, $x_{i}$ are the points in the intervals and $\;\bar{x}$ is the average sample value.

### 4.2. Validation of the Accuracy of the Equipment through a New Case Study

To validate the accuracy of the equipment using the Cloud Compare ICP algorithm, a new comparison was carried out with a bust of a sculpture made in plaster measuring 21.40 cm wide by 33 cm high. The morphological characteristics of the object differ slightly from the previous artifact (vessel), since the sculpture presents different cavities typical of the execution of the work of art. The structure represents the bust of a 42-year-old woman with her hair up (Figure 13).

In the same way and taking the same procedure in the study of the vessel, the integration of several 3D modeling technologies makes the work unprecedented, so a workflow is configured using photogrammetry techniques performed with GCP through control points. The records of the 3D optical measurement systems were carried out through the respective software that is supplied by the equipment manufacturers. All data acquisition parameters were carried out with the same properties and the same equipment, varying the number of photographs of the experimental photogrammetry campaign. Instead of having 142 photographs, 101 images were taken distributed as shown in Figure 14.

These 101 images have been processed using Agisoft Metashape, obtaining a dense cloud with 1.8 million points. This set is called VSfM_v_ which points the software processes to the images through mathematical algorithms on 3D shapes known as Structure from Motion (SfM). As it was, the coded markers work as GCP. Therefore, the scale bars inserted between these coded markers provides a minimal error of 0.0052 mm.

Regarding the scan performed by the Revopoint POP scanner, four tests were carried out. The first, similar to any scanning process, requires the operator’s expertise to capture all surfaces of the object, including oblique areas. The difficulty of this sculpture is the registration of the horizontal parts that comprise from the chin of the face to the neck. The chin area and the part of the hair tail, which are complex surfaces and must be captured from a lower position. Of the four data sets, the last one in which the entire surface was recorded in its entirety was taken. The data sets were identified using a validation ID with the properties and characteristics determined in Table 6.

Once the three data sets VSLS_vs_, VSLS_vp_ and VSfM_v_ have been obtained, we proceed to the automatic focus of point coincidences through the ICP fine record [32]. The process is developed by filtering the cloud of points and sampling the data set. Subsequently, the key points are identified considering the angles and distances between both sets of points. The transformation parameters are calculated automatically by Gauss Markov least squares fitting [51]. The entire method validation process is carried out through the following results. The difference between Artec Spider and Revopoint POP 3D resulted, as shown in Table 7. Figure 15 and Figure 16 show the difference between the point cloud of the scanner Artec Spider and the scanner Revopoint POP.

The difference between Artec Spider and photogrammetry technique resulted, as shown in Table 7. Figure 17 and Figure 18 show the results.

The difference between photogrammetry technique and the scanner Revopoint POP resulted, as shown in Table 7. Figure 19 and Figure 20 show the results.

The distance between subsets of points was calculated by comparing the results of VSLS_VS_, VSLS_VP_ and VSfM_V_. The parameters studied are the root mean square (RMS) error, the minimum and maximum distances between the point clouds, the average distance, the standard deviation and the maximum estimation error. The comparison data is shown in Table 7.

### 4.3. Spatial Resolution of the Point Cloud of the Systems Used

The point density is one of the parameters that determines the quality of the geometry of a 3D mesh. In the modeling process, the number of points and their distribution can determine the morphological parameters of the objects and their final result. The triangular mesh that originates for 3D modeling varies according to the point density parameters [7], modeling algorithms [52,53] and the shape of the object surface [35]. The points density is a parameter that is measured according to the surface area. Therefore, the density of points per area of the point sets was evaluated per square millimeter. The best way to check the spatial resolution of the point cloud, that is, the 3D Euclidean distance (see Equation (3)) between the closest points, is to sample the set of points.
(3)$$d_{E}\left(P_{1,}P_{2},P_{3}\right)=\sqrt{{\left(x_{2}-x_{1}\right)}^{2}+{\left(y_{2}-y_{1}\right)}^{2}+{\left(z_{2}-z_{1}\right)}^{2}}$$
where d_E_ is the Euclidean distance between points in space, and x, y and z are the Cartesian coordinates of those points.

To determine the point density analysis, a sample was taken from a 20 × 20 mm section taken from the flat edge of the vessel according to Figure 21 in the different records analyzed. For the segmentation of the sample, the Cloud Compare software has been used from the alignment of the sets of points. Three subsets of representative points are obtained from a 400 mm^2^. The distribution of points in 3D space is expressed in Figure 22 following the following order: (a) SLS_VS_, (b) SLS_VP_ and (c) SfM_V_.

The subsets of points SLV_VS_ in the 400 mm^2^ area portion obtained a total of 455 points for, which is equivalent to a point density of 1.13 points/mm^2^. For the subset of SLS_VP_ points, a total of 6752 points was obtained, which is equivalent to a density of 16.18 points/mm^2^. Finally, with the photogrammetry technique, a subset of SfM_V_ points of 1465 points has been obtained, which is equivalent to a density of 3.66 points/mm^2^. Figure 23 shows the dispersion error of the subsets of points of the selected points. The points density was measured in square millimeters. 

### 4.4. Accuracy Analysis between Point Clouds Using BIM Environment

Modeling has been considered as a digital representation that contemplates the geometric and simplified properties of a building, archeological site, or object. These models provide an immediate representation of the designed architecture in all three dimensions. When the representation is the total of an object in its envelope and representing the geometric properties, we speak of the 3D model; when it is not complete or partial, we speak of 2.5D. The three-dimensional model (3D) is an important digital form for the registration of heritage documentation [5]. In most cases, the properties of digital models are affected by different acquisition and rendering techniques. Building information modeling (BIM) represents the process of developing and using computer generated modeling to replicate buildings, simulate design, as well as the construction, planning building, and operation of buildings [54]. Currently, few studies have dedicated an effort to bring archeological objects to BIM. The work of Moyano et al. [55] is one of the few studies to establish an approach to a BIM environment. Advances in the BIM digital platform involve the introduction of semantic components, represented as digital objects with relationships, attributes, and properties [56]. In the Cultural Heritage (CH) plan, geometry acquires an essential role, since most buildings have geometries that are more complex than those found in current buildings that are usually built from straight lines. Geometric accuracy is capable of being captured by BIM methodology, and this is a topic discussed by Moyano et al. [57] as a turning point to the question: what are the attributes provided by the point cloud in BIM spaces? The data collection methods facilitate the transfer to BIM environments; thus, the latest versions of the Graphisoft ArchiCAD^®^ software allows for the direct import of point clouds in interoperable formats (.txt, .ptx, .xyz, .e57, .pts), in the same way as Revit. In the case of cultural heritage, the evaluation of the point cloud is in line with structural and geometric forms. The behavior of complex forms and the reproduction of digital replicas are analyzed. In this sense, analyzing the precision between point acquisition models through vector theory in point integration in BIM could be an unprecedented statement in this scientific field, as an evaluation system. The process starts with the integration of two sets of SLS_VS_ and SfM_V_ points in the BIM environment. To do this, and as a test, SLS_VS_ and SfM_V_ are inserted in the BIM ArchiCAD version V 25. The records are georeferenced in a local system set through the point of origin of the project in the BIM environment. The results are shown in Table 8.

For this analysis, different sections are generated to check the correspondence of the shapes in the different x, y, z planes.
(4)$$\vec{a}=a_{1}*\vec{u_{x}}+a_{2}*\;\vec{u_{y}}$$
where $a_{1}$ and $a_{2}$ are the Cartesian components of the vector and:$$\beta =arctg\;\frac{a_{1}}{a_{2}}$$
$$\alpha =arctg\;\frac{a_{2}}{a_{1}}$$
$$arctg=\left(tangent\right)\;{\;}^{-1}$$

For the development of the evaluation, the edges of the outer points of the geometric shapes were detected (Figure 24). In the process, different sets of points represented in the XY plane are generated. This set of points is called Np1 coupled to a vertical Z axis (PE) of the resulting geometry. Np2 is the next set of points created as a comparison element by another section plane. The difference Np1 in Figure 24 can be determined as it is the representation of the profile with the last number of points and that corresponds to the SLSvs. In the same way, Np2 is represented with the profile of points with a higher density.

The result of the analysis leads to taking five vectors in the OXY plane according to Equation (3), corresponding to five different angles in the shape of the development of the geometry of the vessel. The vectors $\vec{a_{1}}$; $\vec{a_{2}}$… are going to establish the maximum and minimum deviations between nearby planes. These measurements are taken through the average marked through guide lines. These auxiliary lines are introduced on the same platform to help parameterize the objects, allowing measurements and data capture. The same guide lines mark the angles that make up these vectors with the vertical Z axis.

Figure 25 shows the insertion of the SLS_VP_ dataset model in the BIM environment, where in the 3D view a system of views and sections can be created through section planes in the reference box of the model to understand the geometry of the model object. The orthophoto projection is inserted into BIM as a worksheet in order to observe the correspondence between the section and its elevation projection.

Figure 26 shows the dispersion of the maximum and minimum deviation expressed in millimeters of the vectors of the section plane generated in a BIM environment of the set of SLS_VS_ and SfM_V_. Distances were extracted in absolute values and with submillimeter precision, configuring the work environment at this scale.

### 4.5. Bring an Archeological Object to BIM

The development of the quality of 3D reconstruction is closely related to the massive data acquisition techniques Massive Data Capture Systems (MDCSs). In turn, the quality of the 3D rendering has a direct correlation with the scan size of the object. Therefore, we will have to distinguish between the scanner registration of a small sculpture to the process that takes the registration of a church or a cathedral, with all its components of floors, mezzanines, and movable objects. The use of appropriate instruments corresponds to the scale on which you are working and the format of its reproduction. On the other hand, 3D models can be generated through different software on the market. For example, the Metashape software itself generates its own 3D mesh or others such as Cloud Compare or MeshLab, which are the two most significant in generating the mesh 3D [7]. To compare the forms of 3D quality of the subsets of previous points in the 400 mm^2^ fraction, geometry has been studied through the generation of the mesh. This means that each 3D representation will originate from a different surface according to the record that has been taken. Thus, by analyzing the projections in the OYZ plane of the planes of the subsets of points, the true analysis of the quality of each record of the SLS_VS_, SLS_VP_ and SfM_V_ point sets can be taken. These three projections are given in Figure 27.

The angle that determines the fraction of points on the surface is expressed in Table 8, and the distance between the marked red lines determines the deviation of the subsets of points.

These analysis values will determine the quality of the 3D surface in the meshing process. Digital surface models (DSM), sometimes called digital terrain models (DTM), represent the elevation of the surface and thus involve the final quality of the object representation. Most scientific studies use the DTM/DSM as city models, forestry applications, structural monitoring, and change detection. There are construction algorithms that reproduce the model [58], although the applicability software itself can modify the parameters and determine that the results show disparate data. A first evaluation carried out with Cloud Compare determines that the mesh is representative of the set of points in origin. For example, in Figure 28 the meshing of the different surfaces of the area of Figure 21 is represented. The Cloud Compare software is based on its octree structure to shape a point cloud spatial coordinate system [59]. The 3D triangular mesh surface modeling is performed using the Poisson Surface Reconstruction algorithm [60]. In this process, the mesh of the set of SLS_VS_ points with a white surface and SfM_V_ with a brown surface obtains positive results. On the other hand, in the subset of SLS_VP_ points that shows a gray wireless surface, the mesh result is distorted. This results in very complex surfaces that are not representative as can be seen in Figure 28 (an error occurs due to alteration of the algorithm).

Subsequently, the SLS_VP_ data set was masked using another software that contains a 3D surface generation algorithm (Figure 29), in this case Rhinoceros 3D V.7 [61], obtaining a representative result of its shape.

The BIM environment, in new construction and Historic Building Information Modelling (HBIM) in heritage areas, defined by Murphy et al. [62], is distinguished from other 3D modeling approaches by the ability to assign semantic data to each parametric object. This is useful in the practices of the AEC (Architecture Engineering Construction) industry, since each object contains information about its classification of construction elements, object identifier, type of renovation cycle, structural loads, among other aspects. Digital tools encourage collaborative work and allow better information management at each stage of the project. Thus, three dimensional digitization is the generation of a 3D computer model [63] where the digital model occupies an essential space to generate images, virtual recreations, and construction models.

Advisory bodies such as BIMFORUM, the British Standards Institution (BSI) or the US Construction Documentation Institute (USIBD), have developed benchmarks to measure the quality and quantity of information, including levels of geometric detail and information associated with different phases of the project. There is also an attempt to address the challenges associated with digitization and measure historic buildings’ deformations. The first studies using TLS combined with the total station for the creation of BIM models in existing buildings were carried out by Tarvo et al. [54], defining the essential details in the process of creating a BIM. The GSA BIM guide for 3D images [64] mentions tolerances for as built deliverables, plans and point clouds. The COBIM guide suggests the accuracy of BIM elements for existing buildings and referring to historical details, a precision of 5 mm, without establishing its size and old irregularities of 50 mm. Bonduel et al. [65] set the guidelines for the Level of Accuracy (LOA) precision levels from level 10 to 50, taking into account the technical guidelines of each country. Tarvo published the precision levels in BIM from USIBD [66] without reiterating that Bonduel had already established them before his publication. Therefore, it must be taken into account that the specific LOA ranges for heritage buildings can be between 0 and 5 cm from a lower to a higher range. Although there are tolerance parameters that determine the aforementioned dimensions in the deliverables as built, planimetry, point cloud acquisition and digital surface models. Currently there are few studies that address points of reference and guidelines applied to cultural heritage. Some are developed for the design and construction of new projects. 

Exposing a new digital surface model evaluation system through a BIM environment means working with adaptive meshes according to the work developed by Moyano et al. [55]. From the same BIM environment as it appears in Figure 30, three meshes are generated representing a digital surface model of the results of the three subsets of segmentations points of the SLS_VS_, SLS_VP_ and SfM_V_. Next, a cyan section plane is drawn, which is a horizontal plane that determines the Z level. In the ArchiCAD environment under the 3D viewer, a custom section plane can be set under an x, y structure. This personalized plane has a slider on the z axis that takes a positive dimension until it reaches the desired level. Figure 30 shows how the digital surface model presents different variabilities showing the different submillimeter surface measurements. The measurements from a graphical point of view can be determined by the different surfaces between the cut plane in cyan color and the relief of the surface.

## 5. Discussion

New image capture technologies based on photogrammetry, as well as structured light scanners, are effective tools as Massive Data Acquisition Systems (MDCS). The use of these systems is essential for 3D digitization studies, providing precise records and without physical contact with the surface, making it ideal for studies of objects of great archeological and architectural value. The research work begins with the data capture records of two scanners; one of low cost and others of high precision, in addition, data are captured using SfM digital photogrammetry techniques. Next, the degree of metric dimensional precision is compared through an Iterative Closest Point (ICP) algorithm. In the applicability of the comparison algorithm, a point adjustment and filtering process are carried out. This way of proceeding ensures that the residual values will not affect the final result of the precision and comparison between systems. The deviations between point clouds, between the two structured light scanners SLS_VS_, SLS_VP_ are very similar, since, in the process of eliminating residual points, the variability in the maximum distances is minimal, going down from 6.7976 to 5.0768 mm as can be verified in the graph of Figure 31.

For the case of the comparison between the SLS_VS_ and SfM_V_ point sets, the deviations reflect greater dispersions than in the previous case, but with a stabilization of the results in the maximum distances. The variability in the maximum distances is insignificant, falling from 10.3225 to 9.2875 with a stop of 9.2875 mm, as can be seen in the graph in Figure 32.

For the case of the comparison between the SfM_V_ and SLS_VP_ point sets, the deviations reflect greater dispersions than in the previous cases. The variability in the maximum distances is very significant, going down from 87.4128 to 29.1707 mm and in the third evaluation 10.12 mm, as can be seen in the graph in Figure 33. In both cases, in (A) with automatic alignment and (B) with manual alignment, the results are very similar. The maximum point spread increases with the photogrammetry system because there are no residual points between the structured light scanners. This is really an advantage that differentiates the structured light scanner from the image capture scanner.

The results of analysis between records of the same object detect that the precision is very similar with variations below a millimeter between the three systems, as can be understood from Table 9 (vessel) and Table 10 (bust).

Table 10 shows the validation of the precision data between the three sets of data obtained in a second experiment from the recording of a woman’s bust. The data obtained from the validation of the precision are very similar, considering that the range oscillates in the first case (Spider_SfM) a difference of 0.212 mm. In the second case (Spider-POP) of 0.551 mm and lastly in the third case (SfM-POP) of only 0.04 mm in absolute value.

In photogrammetry, processing occupies an essential space to determine the density of points. These parameters determine the quality of the cloud, triangle meshes, and texturing. In the case of the Artec Spider structured light scanner, the embedded software also facilitates different levels of quality. Thus, the quality of the point cloud also deter-mines the quality of the mesh [67] and therefore the reconstruction of the 3D model. In the case of the Revopoint POP 3D Scanner, the parameters are set by default in the software, so there are no quality parameters. The density of points intervenes not only in the quality of the final result of the 3D digitization, but also varies the processing algorithm chosen. For the study of density and the evaluation of the mesh, an area of 400 mm^2^ was segmented from the edge of the vessel was segmented. The results show that the Revopoint POP 3D Scanner obtains the highest number of points per surface unit, followed by photogrammetry and lastly the Artec Spider scanner. The distribution of points in the OYZ projection plane according to Figure 25 determines a variation in points where the Artec Spider scanner behaves with the best results of Alpha angle tilt and distribution of points. In fact, the survey point density box plots on the segmentation of the representative point subset of the rim plane of the vessel shown in the plot in Figure 21, represent the Artec Spider scanner as the most robust.

On the other hand, in this work, it was experimented on how the different point clouds influence as records acquired from the three systems in the evaluation of the mesh for micro surfaces. The subsets of points from the SLS_VS_, SLS_VP_ and SfM_V_ records were processed using the Poisson Surface Reconstruction algorithm for the calculation of 3D triangular mesh surface modeling of the C2C software, for implementation in a BIM environment. This process can be operational: (i) once the mesh is obtained, it is automatically adapted to a GDL object and inserted into the ArchiCAD library. The critical component of this part of the workflow requires rotating the objects so that the mesh surface is as aligned as possible in the (x,y) plane; (ii) the GDL object must be transformed into a morph element; (iii) from the construction of a regular parametric solid—Wall, Slab, Column or Mesh—and through Boolean operations, the adaptability of the surface defined by the point cloud to that parametric element is reached according to the mathematical parameters of the expression 4. Finally, (iv) the definition of the real shape of the segmented objects was used as a division surface (subtraction with upward extrusion in ArchiCAD software. This process is called as the reconstruction of the model by the adaptive mesh.
(5)$$\{\begin{array}{c} T{\;}_{\underset{u}{\to }\;}:\;R^{n}\to R^{n}\\ P\;\to P^{\prime }=T\left(P\right)=P+\vec{u}\; \end{array}\;\;\;\;\;\;\vec{{PP}^{\prime }}\;=\;\vec{u}$$

The production of meshes to BIM to reach BIM digital environment platforms requires significant conversion, workflow, and operator knowledge. 

Based on the generation of a digital surface model of the results of the three point subsets of the SLS_VS_, SLS_VP_ and SfM_V_ segmentation in the ArchiCAD environment under the 3D viewer, a custom cutting plane (x,y). This custom plane determines areas of surfaces that are distinct and whose sweeps have submillimeter values. Taking into account this morphological structure of the subsets of points, it must be said that the DSM varies according to the structure of the density of points obtained from the records of the three systems analyzed. 

The great versatility of BIM platforms makes it possible to insert a set of exported point clouds in different formats for later analysis. In this way, a vector theory was created in BIM that determines the maximum and minimum deviation expressed in millimeters of the vectors of the section plane generated from the set of SLS_VS_ and SfM_V_ points. The distances were extracted in absolute values and with submillimeter precision, configuring the work environment at this scale. The results that are expressed in the graph of Figure 24 show that at angles of 188° (1.9830 mm) the dispersion is greater, the points that are in the perpendicularity being the most solid, reaching even values of tenths of millimeters. (0.0100 mm). In a model where the variations between records are found in the expressed measurements, it can be said that there are minimal differences, and these can be tolerated to generate a model or a digital replica.

## 6. Conclusions

With a view to evaluate the accuracy of the low-cost and high-cost data capture technologies addressed in this research, a mathematical algorithm is used on segmented point sets so that their spatial resolution is verified. In addition, the precision of the 3D model is analyzed through a profile analysis using vectors. The results reveal similar geometrical data and the variables to be considered in the different techniques used. Photogrammetry achieved an accuracy of 0.70 mm versus the Artec Spider and 0.57 mm against the Revopoint POP 3D scanner. Therefore, the difference of 0.93 mm between the two structured light scanners makes this variation in accuracy acceptable to operators and researchers in archeology and historical architecture, who could use low-cost structured light scanning. The comparison of the different devices and technologies allows for choosing the right means to accurately capture the geometry of the assets, creating predictive degradation models, evaluating sculpture painting, and detecting pathologies related to stone mineralogy. This research also shows that digital twins of assets from 3D scanning and SfM point clouds, as a parametric building information model (BIM), permit their evaluation and the representation of complex shapes in the artistic sculpture domain. Although the evaluation procedure in a BIM environment is unprecedented, the reference framework in benchmarking research uses comparison algorithms based on software such as CloudCompare or Geomatic. The scientific community acknowledges that advanced digital tools allow better information management at each stage of the project. Although this may be used in the future to establish precision models in 3D digital reconstruction, a crucial topic in the fields of archeology and architecture. In reference to all of the above and, taking into account the development of commercially available instruments, there are numerous investigations that focus their work on the use of small-dimensional scanners for the massive use of records of pieces from museums and archaeological sites. The value of this work lies in covering the knowledge gap about the applicability to the precision demand of low-cost 3D optical sensors. In addition, this document provides the evaluation of said 3D sensors under a BIM Platform environment through vector analysis, an analysis unpublished up to now.

Finally, the test results obtained are only valid for the specific model of each system investigated, that is, the accuracies achieved do not constitute a universal statement about the specific portable 3D scanning systems analyzed. Further work is still needed, such as the analysis of texture quality and color perception for the segmentation of integral parts of works of art. Nevertheless, the authors believe that this research paper can lead to theories about the accuracy of 3D digital reconstruction models. In this line, at the work planning stage, DIGITALEUROPE stresses the need to have a digital version of the project before it is built, and that the physical artifact should be a digital twin [68]. Likewise, proposing a BIM-based vector theory to analyze individual fragments of complex surfaces will be necessary and advisable.

## Figures and Tables

**Figure 1 sensors-23-01730-f001:**
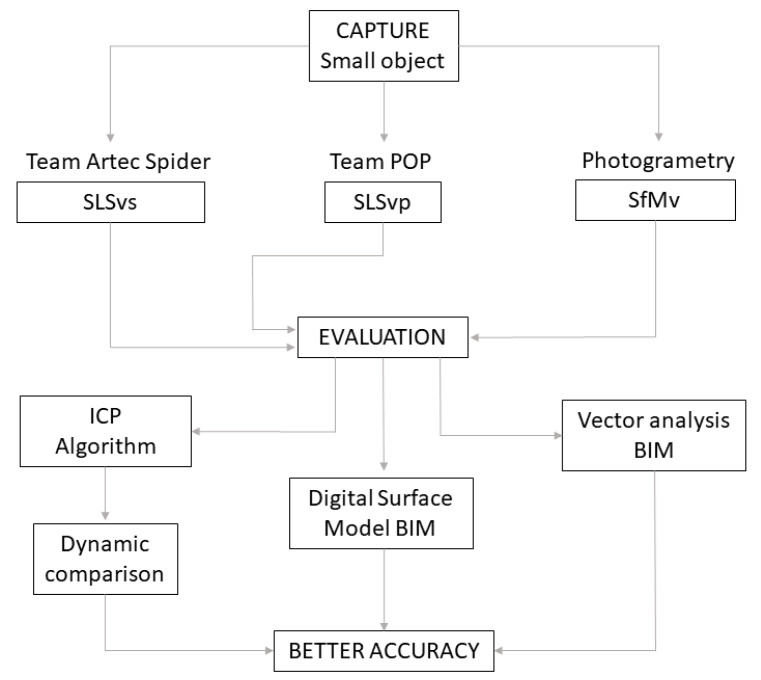
Experimentation workflow description.

**Figure 2 sensors-23-01730-f002:**
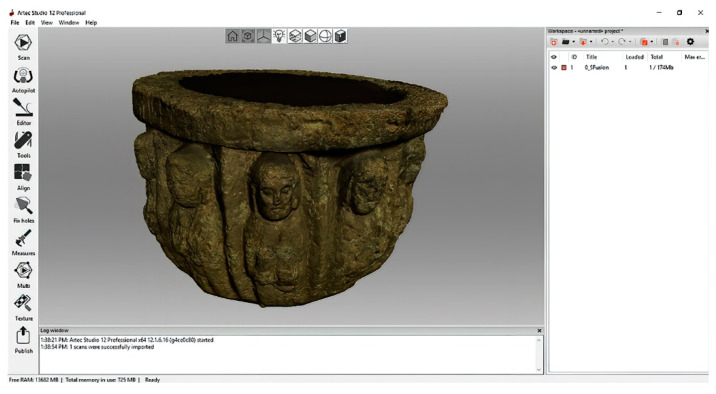
Capture process using Artec Studio Professional v.12 software.

**Figure 3 sensors-23-01730-f003:**
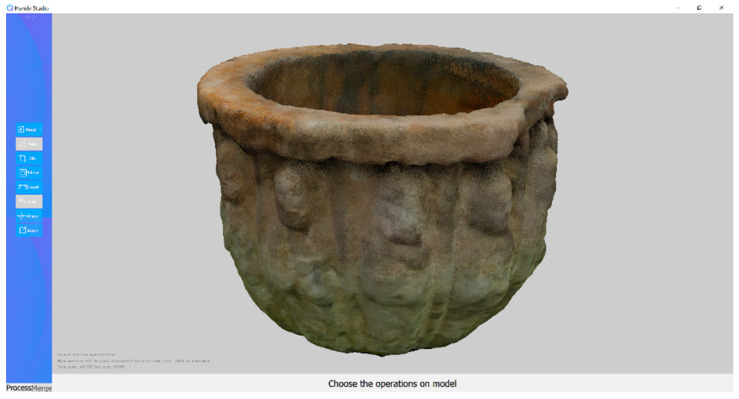
Capture process using HandySCAN software.

**Figure 4 sensors-23-01730-f004:**
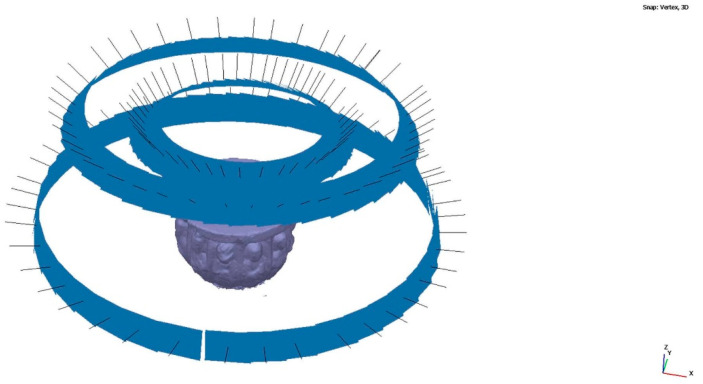
Process of capturing images in photogrammetry.

**Figure 5 sensors-23-01730-f005:**
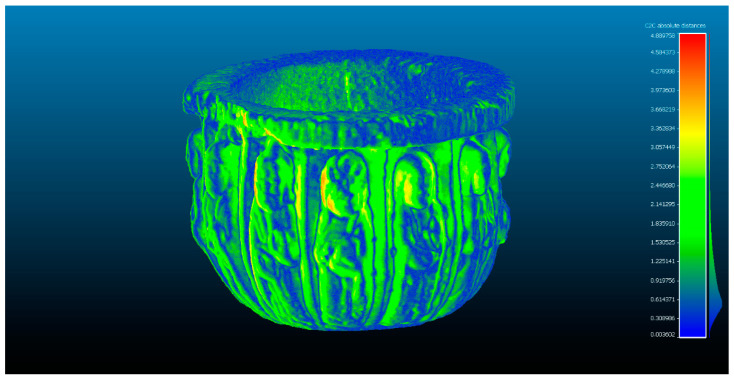
Analysis of the difference between the point cloud of the scanners Artec Spider and Revopoint POP in evaluation 2. Result of applying the C2C algorithm. Color map of the distribution of distances. On the color scale, red represents the maximum distance expressed in millimeters.

**Figure 6 sensors-23-01730-f006:**
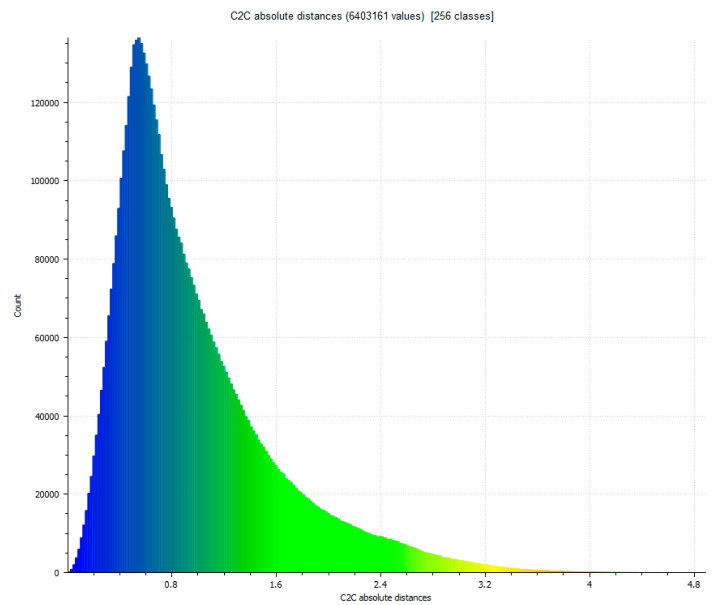
Histogram of Figure 5. Histogram units: millimeters (X axis) and number of points (Y axis).

**Figure 7 sensors-23-01730-f007:**
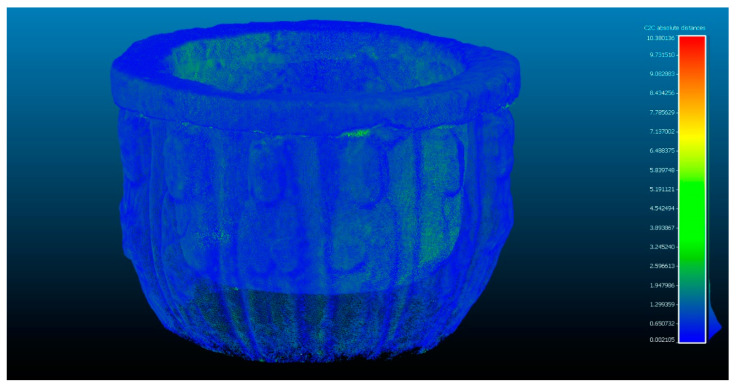
Analysis of the difference between the point cloud of the scanner Artec Spider and SfM in evaluation 3. Result of applying the C2C algorithm. Color map of the distribution of distances. On the color scale, red represents the maximum distance expressed in millimeters.

**Figure 8 sensors-23-01730-f008:**
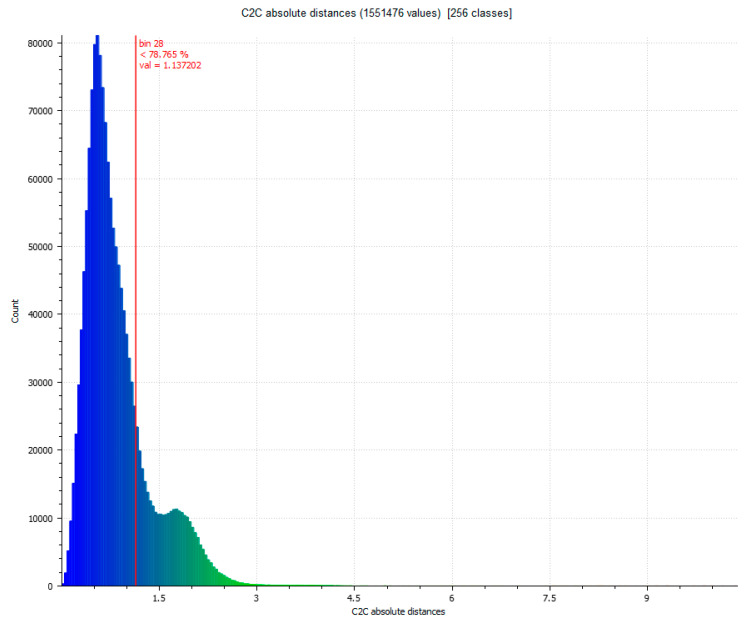
Histogram of Figure 7. Histogram units: millimeters (X axis) and number of points (Y axis).

**Figure 9 sensors-23-01730-f009:**
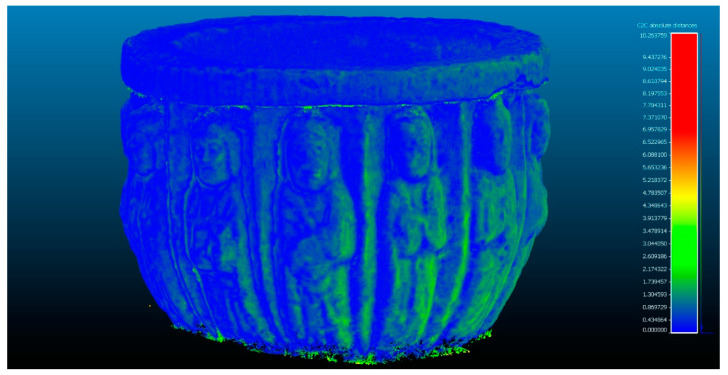
Analysis of the difference between the point cloud (Phase 1) between photogrammetry technique and the scanner Revopoint POP in evaluation 3. Result of applying the C2C algorithm. Color map of the distribution of distances. On the color scale, red represents the maximum distance expressed in millimeters.

**Figure 10 sensors-23-01730-f010:**
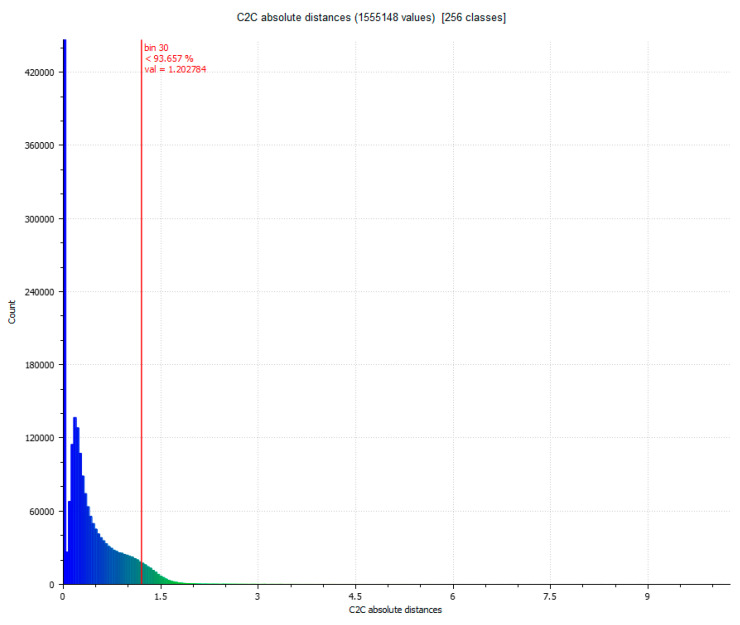
Histogram of Figure 9. Histogram units: millimeters (X axis) and number of points (Y axis).

**Figure 11 sensors-23-01730-f011:**
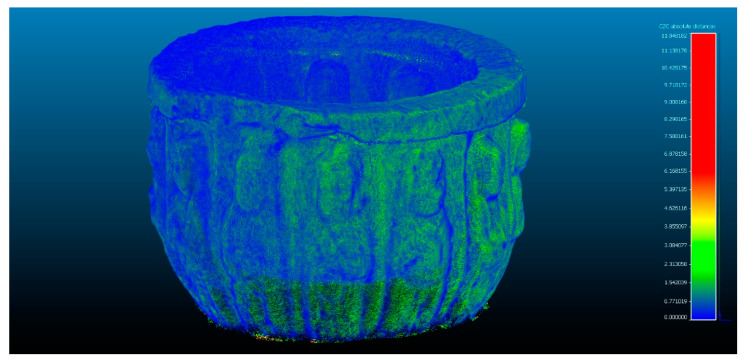
Analysis of the difference between the point cloud (Phase 2) between photogrammetry technique and the scanner Revopoint POP in evaluation 3. Result of applying the C2C algorithm. Color map of the distribution of distances. On the color scale, red represents the maximum distance expressed in millimeters.

**Figure 12 sensors-23-01730-f012:**
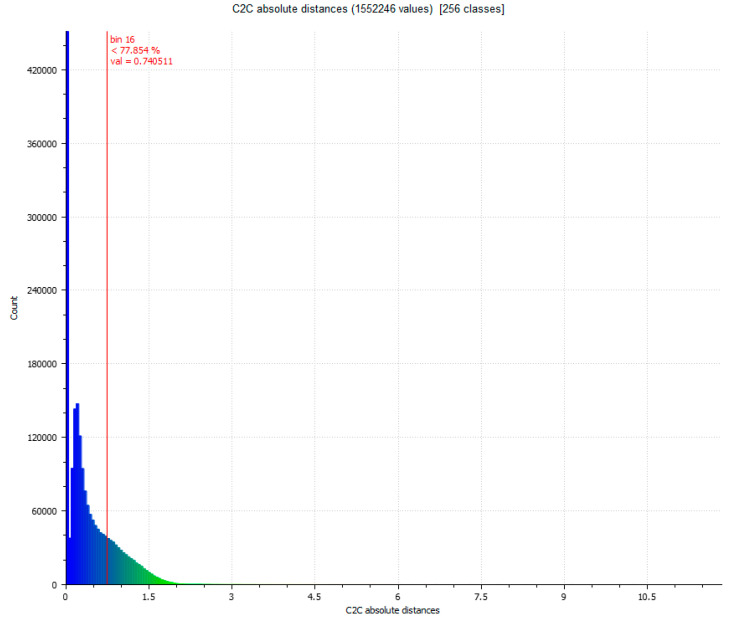
Histogram of Figure 11. Histogram units: millimeters (X axis) and number of points (Y axis).

**Figure 13 sensors-23-01730-f013:**
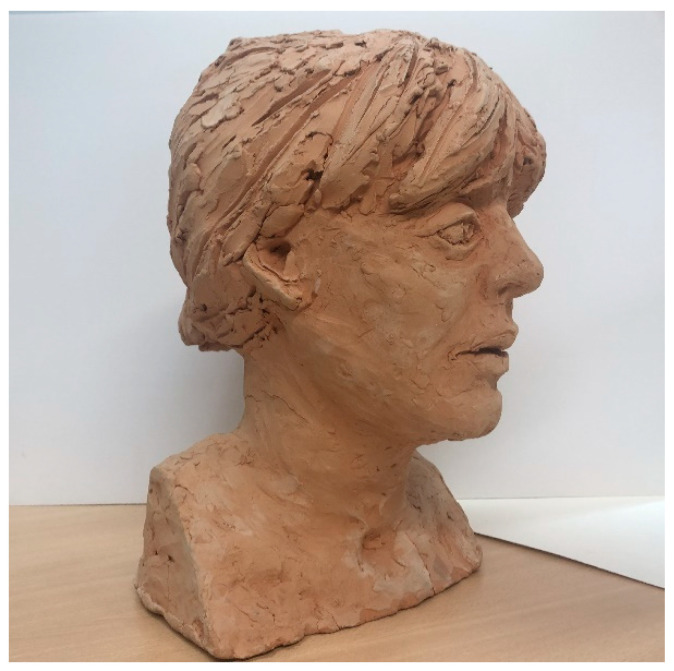
Woman bust image.

**Figure 14 sensors-23-01730-f014:**
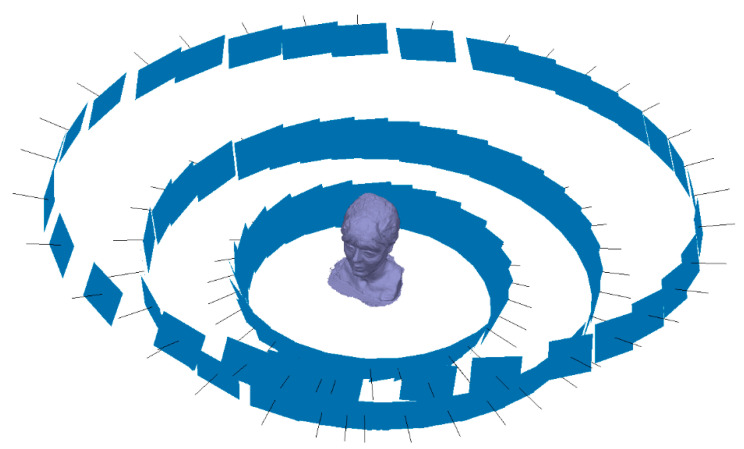
Process of capturing images in photogrammetry of Figure 13.

**Figure 15 sensors-23-01730-f015:**
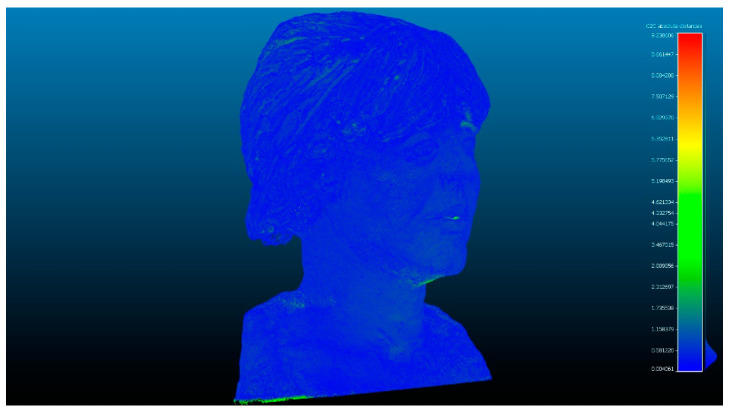
Analysis of the difference between the point cloud of the scanner Artec Spider and the scanner Revopoint POP. Result of applying the C2C algorithm. Color map of the distribution of distances. On the color scale, red represents the maximum distance expressed in millimeters.

**Figure 16 sensors-23-01730-f016:**
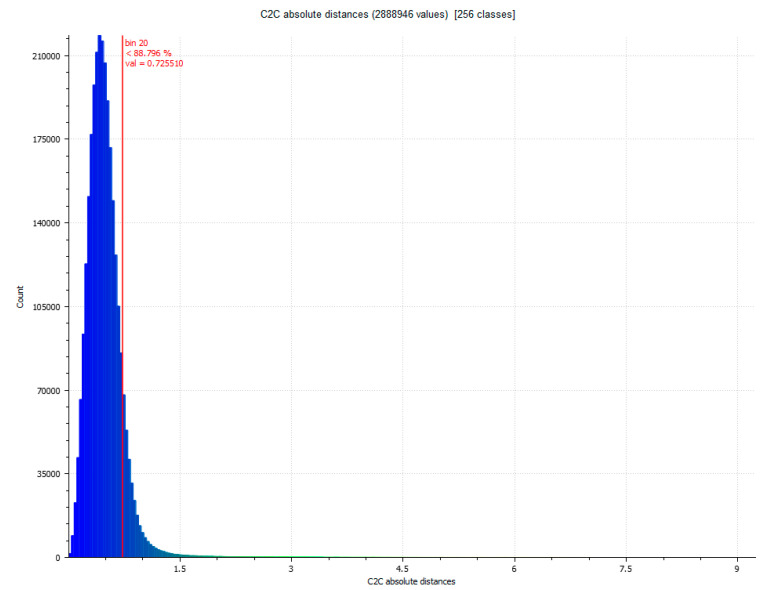
Histogram of Figure 15. Histogram units: millimeters (X axis) and number of points (Y axis).

**Figure 17 sensors-23-01730-f017:**
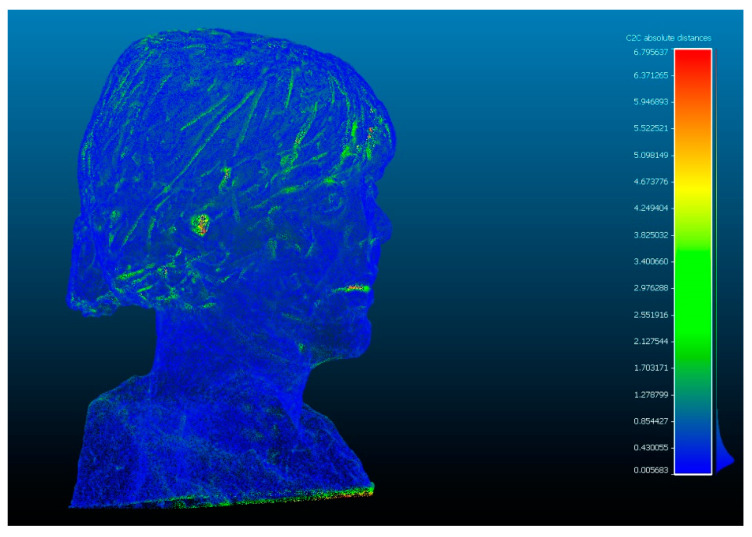
Analysis of the difference between the point cloud of the scanner Artec Spider and SfM. Result of applying the C2C algorithm. Color map of the distribution of distances. On the color scale, red represents the maximum distance expressed in millimeters.

**Figure 18 sensors-23-01730-f018:**
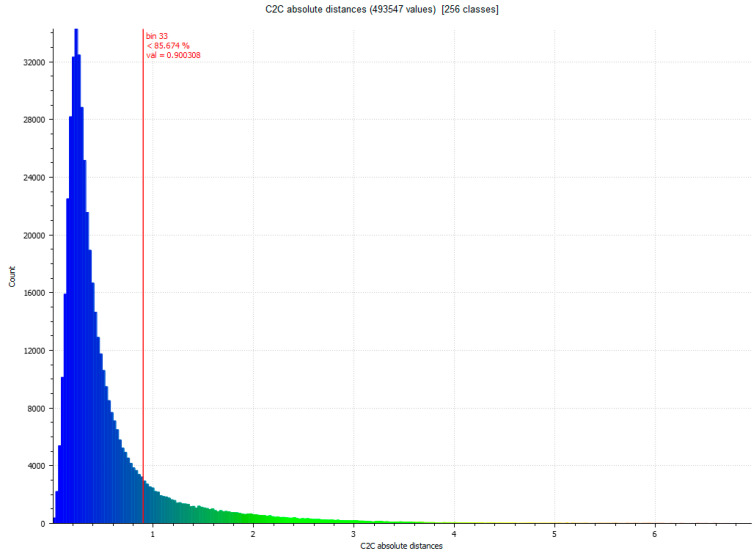
Histogram of Figure 17. Histogram units: millimeters (X axis) and number of points (Y axis).

**Figure 19 sensors-23-01730-f019:**
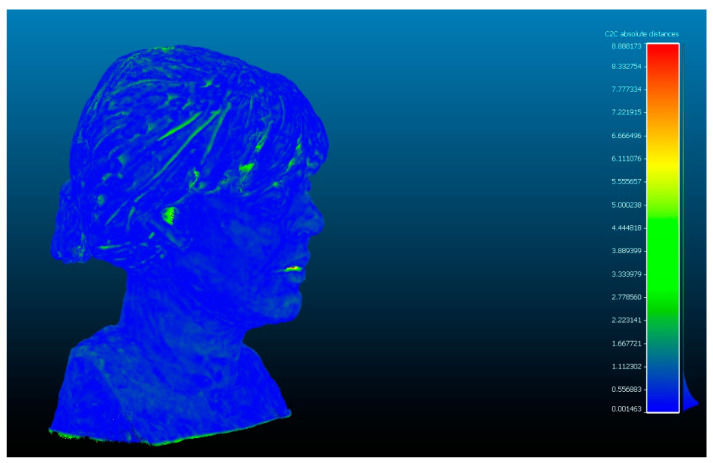
Analysis of the difference between the point cloud between photogrammetry technique and the scanner Revopoint POP. Result of applying the C2C algorithm. Color map of the distribution of distances. On the color scale, red represents the maximum distance expressed in millimeters.

**Figure 20 sensors-23-01730-f020:**
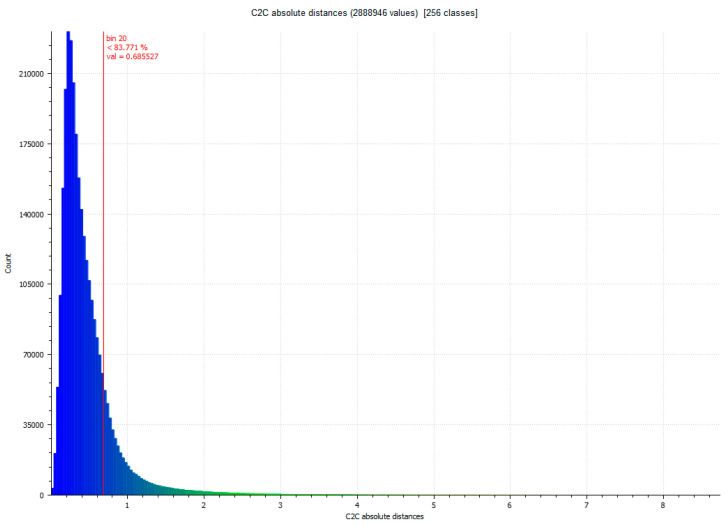
Histogram of Figure 19. Histogram units: millimeters (X axis) and number of points (Y axis).

**Figure 21 sensors-23-01730-f021:**
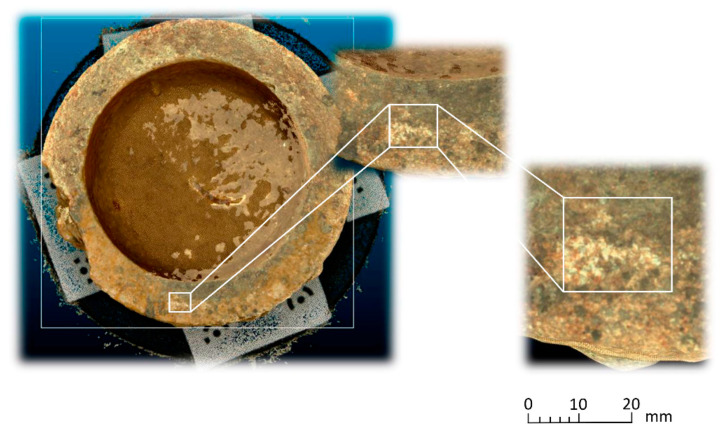
20 × 20 mm segment, it is a subset of points coming from the edge of the vessel.

**Figure 22 sensors-23-01730-f022:**
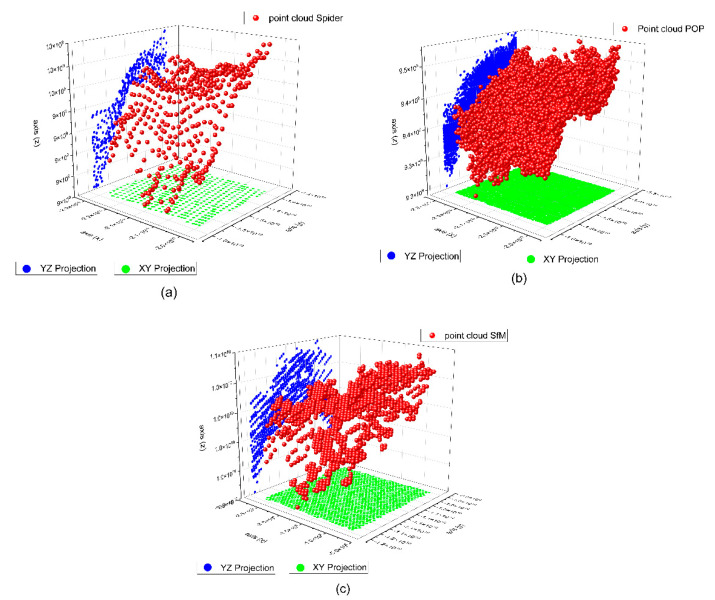
Distribution of subsets of representative points in the rim plane of the vessel. (**a**) Points registered with the scanner Artec Spider, (**b**) points registered with the scanner Revopoint POP and (**c**) points registered by photogrammetry.

**Figure 23 sensors-23-01730-f023:**
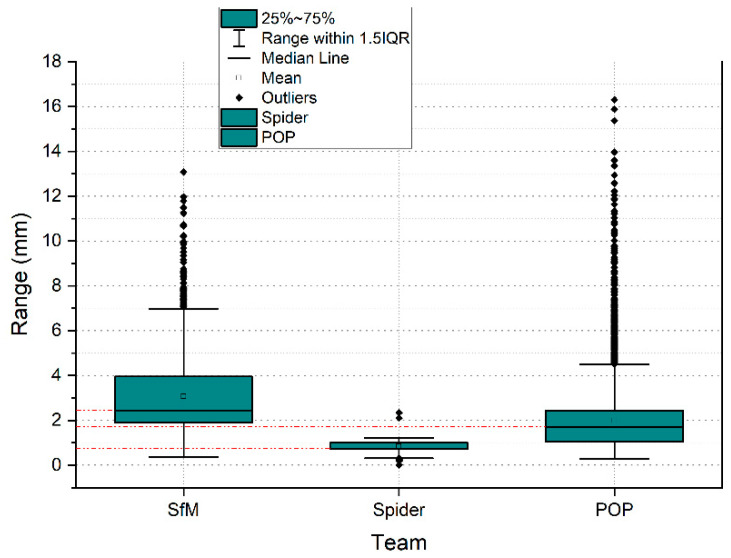
Box plots of the density of survey points in the segmentation of the subset of representative points of the plane of the edge of the vessel.

**Figure 24 sensors-23-01730-f024:**
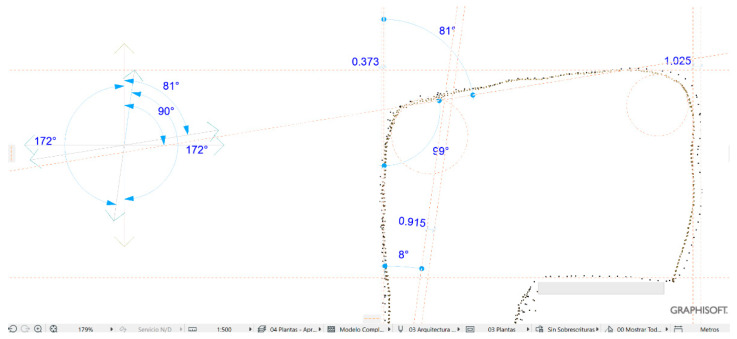
Point cloud model using BIM geometry.

**Figure 25 sensors-23-01730-f025:**
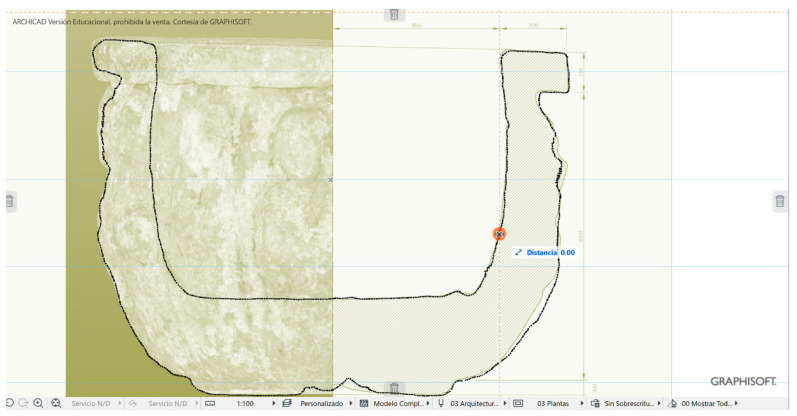
Image of the apostolic vessel in the BIM space with the insertion of SLS_VP_.

**Figure 26 sensors-23-01730-f026:**
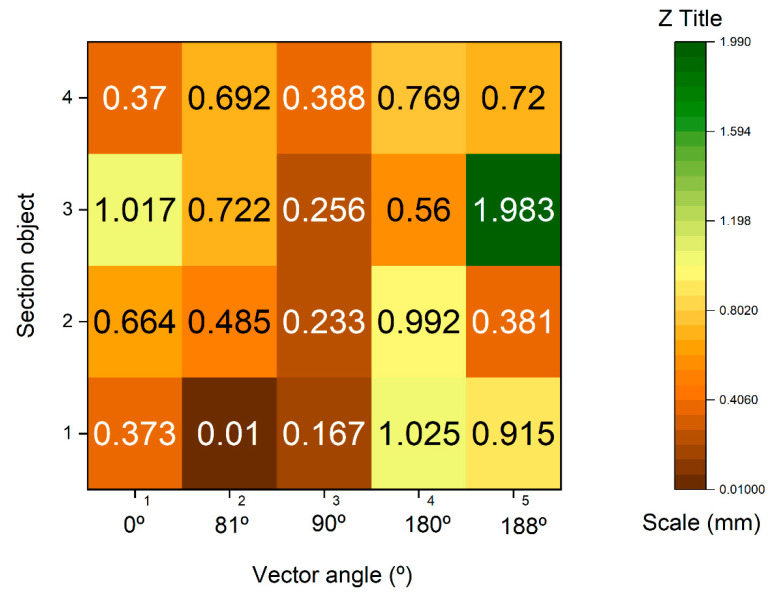
Maximum and minimum deviation dispersion expressed in millimeters of the section plane vectors in the BIM platform.

**Figure 27 sensors-23-01730-f027:**
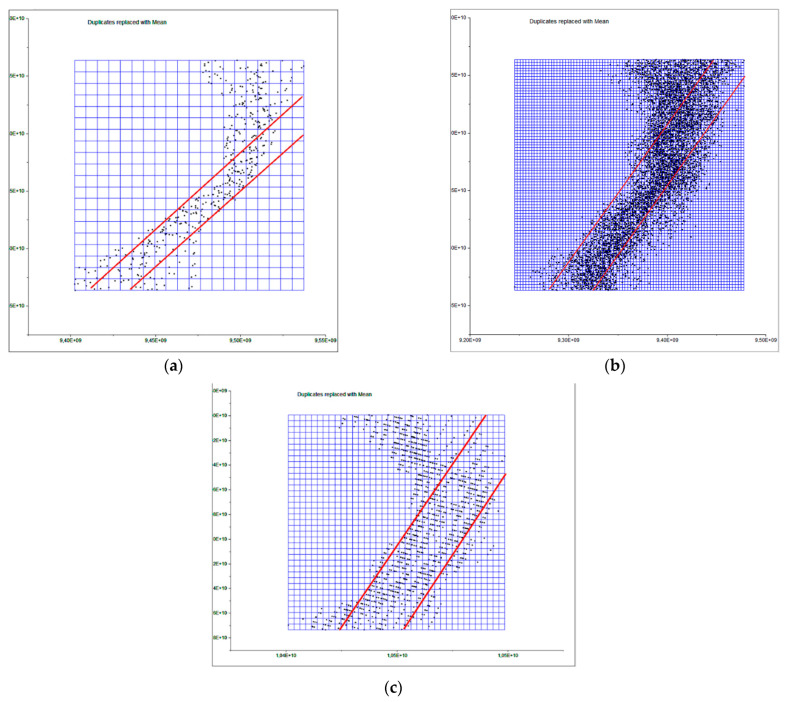
Representation of the subsets of points in the OYZ projection plane, (**a**) SLS_VS_, (**b**) SLS_VP_ and (**c**) SfM_V_. The abscissa axis represents the y-axis, and the ordinate axis represents the z-axis.

**Figure 28 sensors-23-01730-f028:**
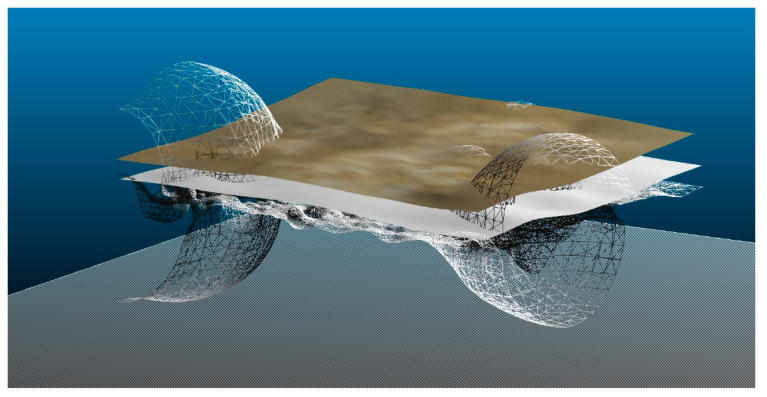
Three-dimensional mesh representation of the surface of the SLS_VS_, SLS_VP_ and SfM_V_ points subsets.

**Figure 29 sensors-23-01730-f029:**
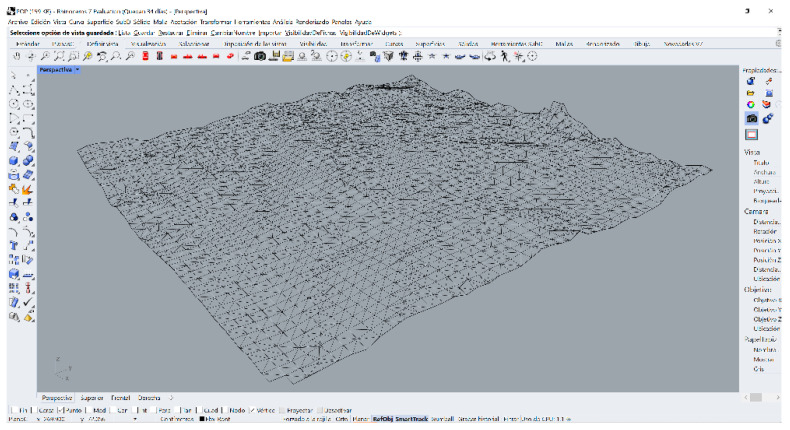
Three-dimensional mesh rendering in Rhinoceros 7.

**Figure 30 sensors-23-01730-f030:**
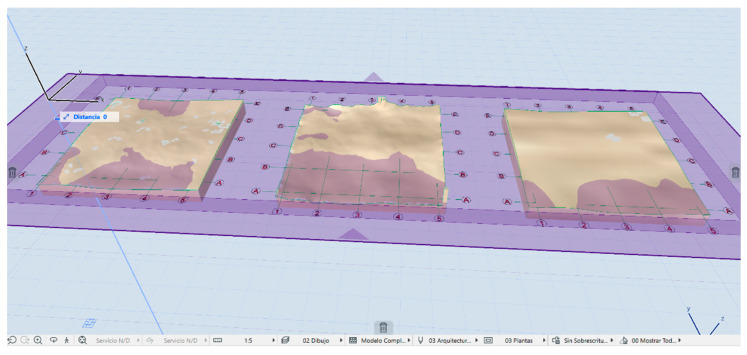
Representation of a digital surface model of the results of the three subsets of segmentation points of the SLS_VS_, SLS_VP_ and SfM_V_.

**Figure 31 sensors-23-01730-f031:**
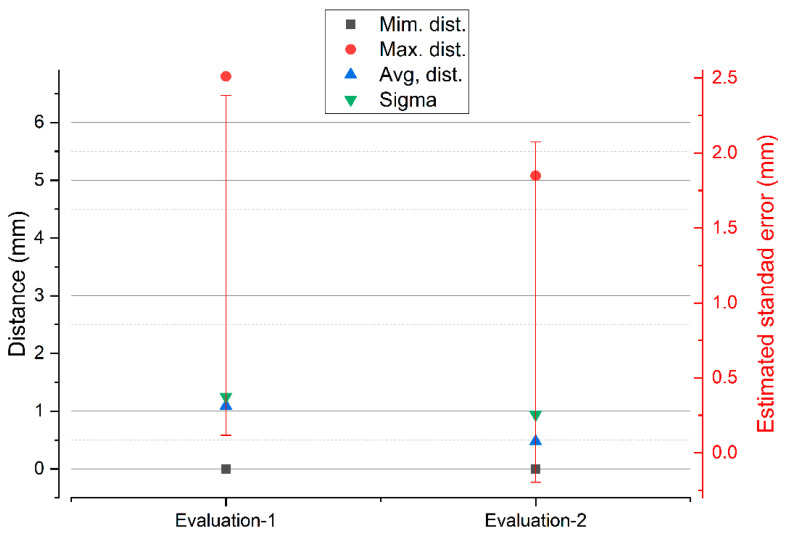
Results of the two consecutive evaluations of segmentation in the set of points between SLS_VS_ and SLS_VP_.

**Figure 32 sensors-23-01730-f032:**
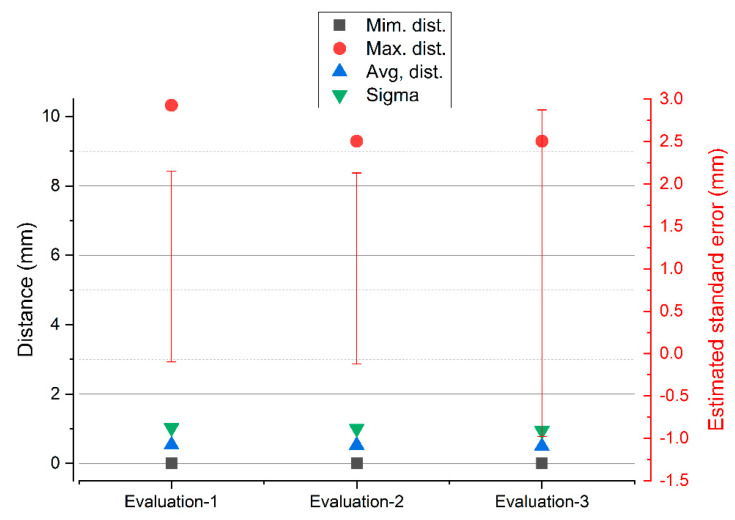
Results of the three consecutive evaluations of segmentation in the set of SLS_VS_ and SfM_V_ points.

**Figure 33 sensors-23-01730-f033:**
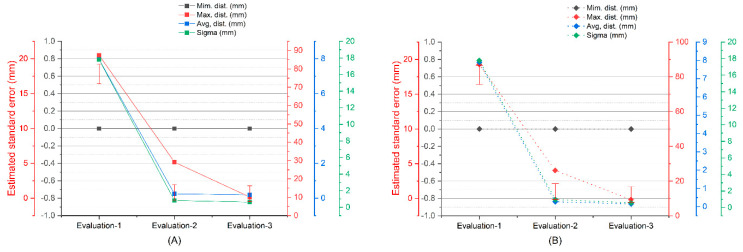
Results of the three consecutive evaluations of segmentation in the set of points between SfM_V_ and SLS_VP_, (**A**) alienation automatic (**B**) alienation manual.

**Table 1 sensors-23-01730-t001:** Main specifications of the camera data acquisition.

Panasonic DMC-GF3	
Nº of images	142
Resolution	12 MP
Distance to the object	≤0.50 m
ISO	320
Sensor	Live MOS (17.3 × 13 mm)
Exposure	1/60 s f 3.5

**Table 2 sensors-23-01730-t002:** Processing setup “Align Photos” and “Build Mesh”.

Step	Parameter	Selection
Align cameras	Accuracy	High
	Generic/Reference preselection	Yes
	Key point limit	40,000
	Tie point limit	4000
	Adaptive camera model fitting	Yes
Build dense cloud	Quality	High
	Filtering mode	Moderate
	Calculate point colors	Yes
Build mesh	Source data	Dense cloud
	Quality	Medium
	Surface type	Arbitrary

**Table 3 sensors-23-01730-t003:** Comparison between Spider and POP.

Comparison between SPIDER and POP	Standard Deviation (σ) (mm)	Min. Distance (mm)	Max. Distance (mm)	Average Distance (mm)	Estimated Standard Error (mm)
Evaluation 1	1.2863	0	7.1653	1.2533	1.1352
Evaluation 2	1.0338	0	5.5619	0.6260	1.1352

**Table 4 sensors-23-01730-t004:** Comparison between Spider and SfM.

Comparison between SPIDER and SfM	Standard Deviation (σ) (mm)	Min. Distance (mm)	Max. Distance (mm)	Average Distance (mm)	Estimated Standard Error (mm)
Evaluation 1	1.0269	0	10.3225	0.5361	1.1263
Evaluation 2	1.0038	0	9.2875	0.5259	1.1263
Evaluation 3	0.9475	0	9.2875	0.4952	1.1263

**Table 5 sensors-23-01730-t005:** Comparison between SfM and POP.

Comparison between SfM and POP	Standard Deviation (σ) (mm)	Min. Distance (mm)	Max. Distance (mm)	Average Distance (mm)	Estimated Standard Error (mm)
Automatic alignments with 6 points					
Evaluation 1	17.7964	0	87.4128	7.9585	1.4026
Evaluation 2	0.8249	0	29.1707	0.2519	1.1320
Evaluation 3	0.6299	0	10.1253	0.1871	1.1320
Best fitting manual					
Evaluation 1	17.7819	0	87.1341	7.8895	1.3875
Evaluation 2	0.9804	0	26.1719	0.2729	1.1305
Evaluation 3	0.5731	0	9.3219	0.1535	1.1305

**Table 6 sensors-23-01730-t006:** Identification of the data set extracted from the equipment.

	Dataset ID	Number of Point	Output File	Scale	Number of Segment Points	Points Density (pto./mm^2^)
Artec Spider	VSLV_Vs_	495.964	.stl	mm	1.268	2.776
POP 3D	VSLS_VP_	2.982.902	.ply	mm	6.592	18.593
SfM	VSfM_V_	1.653.479	.e57	m	2.999	7.028

**Table 7 sensors-23-01730-t007:** Results of a second survey to validate accuracy.

Comparison between	Standard Deviation (σ) (mm)	Min. Distance (mm)	Max. Distance (mm)	Average Distance (mm)	Estimated Standard Error (mm)
VSfM_V_ and VSLS_VP_	0.6183	0	8.5811	0.1480	1.2936
VSLV_VS_ and VSLS_VP_	0.4830	0	9.7524	0.0901	1.3032
VSLV_VS_ and VSfM_V_	0.7350	0	8.2376	0.2097	1.3025

**Table 8 sensors-23-01730-t008:** Determination of the angle and distance measures of the subsets of points in the projections on the OYZ plane.

	α Dispersion Angle	Distance between Lines (mm)
SLV_VS_ (SPIDER)	42°	3.5
SLV_VP_ (POP)	57°	9
SfM_V_ (SFM)	54°	5

**Table 9 sensors-23-01730-t009:** Evaluation of standard deviation in mm between the different records (vessel).

	Spider-SFM	Spider-POP	SFM-POP
Standard deviation (σ) (mm)	0.9475	1.0338	0.5731

**Table 10 sensors-23-01730-t010:** Evaluation of standard deviation in mm between the different records (bust).

	Spider-SFM	Spider-POP	SFM-POP
Standard deviation (σ) (mm)	0.7350	0.4830	0.6183

## Data Availability

Not applicable.

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
