# Peer review of "Evaluation of Geometric Data Registration of Small Objects from Non-Invasive Techniques: Applicability to the HBIM Field"

_sensors, 2023, doi:10.3390/s23031730_

Round 1

Reviewer 1 Report (New Reviewer)

 The topic is interesting and may offer values for a research article related to sensors; however, the manuscript lacks a smooth flow and is not easy to follow because most of the provided information seems disorganized. In addition, the manuscript is not well-organized in terms of providing concise divisions; e.g,. section 3 is very lengthy and repetitive to some extent. Also, most of the figures are not of high quality. This reviewer recommends the following comments to enhance the quality and readability of the manuscript:

The methodology of this research is not well-developed and easy to follow. It looks like that the sample study (a small artifact that is used as a case study in this research) is mixed with the methodology and makes the methodology long and sophisticated. The others may need to separate the case study and the methodology and explain each of them in a proper manner.

The authors may need to enhance their literature review (related work section) by adding some of the most recent research studies on this topic. Although the authors has introduced a few, it does not seem sufficient.

There is not sufficient focus on the sensors part. Given the fact that the authors submit their manuscript to the “Sensors” journal, it is expected that the authors add detailed explanations regarding the sensors used for data acquisition along with the sensors’ characteristics.

Section 3 (Methodology) is overall very lengthy. The authors need to work on it meticulously to significantly reduce the size of this section and offer a concise writing.

Sections numbering is incorrect. Section 3 is “Methodology” and again Section 3 is “Discussion.” The authors should thoroughly review the entire manuscript to correct these types of errors.

The authors may also need to further elaborate on the suitability of the sensors used in relation to their specific research objective. This part is lacking the manuscript to some degree and needs to get fixed.

Most of the figures in this manuscript lacks the sufficient quality and readability. The authors should provide high-quality versions of all the shown figures.

Author Response

Response to Reviewer 1 Comments

The topic is interesting and may offer values for a research article related to sensors; however, the manuscript lacks a smooth flow and is not easy to follow because most of the provided information seems disorganized. In addition, the manuscript is not well-organized in terms of providing concise divisions; e.g,. section 3 is very lengthy and repetitive to some extent. Also, most of the figures are not of high quality. This reviewer recommends the following comments to enhance the quality and readability of the manuscript.

We appreciate the Reviewer’s suggestion. We thank the anonymous Reviewer for her/his time spent to suggest improvements for our paper.

Point 1: The methodology of this research is not well-developed and easy to follow. It looks like that the sample study (a small artifact that is used as a case study in this research) is mixed with the methodology and makes the methodology long and sophisticated. The others may need to separate the case study and the methodology and explain each of them in a proper manner.

Response 1: We appreciate the Reviewer’s suggestion. The chapters of the paper have been restructured, modifying and providing a new structure according to the reviewer's instructions.

Point 2: The authors may need to enhance their literature review (related work section) by adding some of the most recent research studies on this topic. Although the authors has introduced a few, it does not seem sufficient.

Response 2: We appreciate the Reviewer’s suggestion. the review of the literature has been updated in its most relevant and up-to-date aspects. “Thus, in the field of studying architectural heritage, the integration of information from multiple sensors is also studied, combining metric data with temperatures and the creation of a 3D thermal texture [18,19]. Or the analysis of the texture of the color and 3D shape [20]. And the changes in the deformations of the Badillo et al. [21] paintings.”

Point 3: There is not sufficient focus on the sensors part. Given the fact that the authors submit their manuscript to the “Sensors” journal, it is expected that the authors add detailed explanations regarding the sensors used for data acquisition along with the sensors’ characteristics.

Response 3: We appreciate the Reviewer’s suggestion. “The measuring range sensors are between 0.17 and 0.35 m. and data acquisition speed, up to 1 min points/seg. The 3D shape measurement system setup and system calibrations has been verified by Zhao et al. [32].” comment added to the paper and the last sentence can give the reader an idea about the characteristics of these types of sensors.

Point 4: Section 3 (Methodology) is overall very lengthy. The authors need to work on it meticulously to significantly reduce the size of this section and offer a concise writing.

Response 4: We appreciate the Reviewer’s suggestion. The authors have restructured section 3 of the methodology.

Point 5: Sections numbering is incorrect. Section 3 is “Methodology” and again Section 3 is “Discussion.” The authors should thoroughly review the entire manuscript to correct these types of errors.

Response 5: We appreciate the Reviewer’s suggestion. Indeed, there is an error in the numbering that has already been corrected.

Point 6: The authors may also need to further elaborate on the suitability of the sensors used in relation to their specific research objective. This part is lacking the manuscript to some degree and needs to get fixed.

Response 6: We appreciate the Reviewer’s suggestion.

Point 7: Most of the figures in this manuscript lacks the sufficient quality and readability. The authors should provide high-quality versions of all the shown figures.

Response 7: We appreciate the Reviewer’s suggestion. Figures have been improved. Most of the figures have been corrected.

Reviewer 2 Report (New Reviewer)

Author Response

Response to Reviewer 2 Comments

The writing and structure of this manuscript are suitable as a whole. However, several statements could be rewritten more clearly than present form.

We appreciate the Reviewer’s suggestion. We thank the anonymous Reviewer for her/his time spent to suggest improvements for our paper.

Point 1: 

For example, “The process is developed with a filtering of the cloud of points and the sampling of the data set.” An edited form: The process is developed by filtering the cloud of points and sampling the data set.

as shown in table 7. An edited form: as shown in Table 7

The following sentence should be checked for punctuation: “These analysis values will determine the quality of the 3D surface in your meshing process. Digital surface models (DSM) sometimes called digital terrain model (DTM) represent the elevation of the surface and thus involve the final quality of the object representation.” An edited form: These analysis values will determine the quality of the 3D surface in your meshing process. Digital surface models (DSM), sometimes called digital terrain models (DTM), represent the elevation of the surface and thus involve the final quality of the object representation.

This sentence may be unclear or hard to follow: “There is also an attempt to address the challenges associated with digitization and to measure the deformations of historic buildings.” An edited form: There is also an attempt to address the challenges associated with digitization and measure historic buildings' deformations.

This sentence could be rewritten more clearly than present form. It appears that obtained may be unnecessary in the second part. “Table 10 shows the validation of the precision data between the three sets of data obtained in a second experiment obtained from the recording of a woman's bust.”

Please improve the readability of the following sentence: “Finally, it should be noted that the test results achieved are only valid for the specific model of each investigated system, that is, the accuracies achieved”

This sentence may be hard to follow: “Likewise, it will be necessary and advisable to propose a BIM-based vector theory to analyze individual fragments of complex surfaces.” An edited form: Likewise, proposing a BIM-based vector theory to analyze individual fragments of complex surfaces will be necessary and advisable.

Response 1: We appreciate the Reviewer’s suggestion. The sentences have been edited.

Point 2: Please define HBIM in the sentence, double-check the whole manuscript, and ensure all acronyms are mentioned.

Response 2: We appreciate the Reviewer’s suggestion. “The BIM environment, in new construction and Historic Building Information Modelling (HBIM) in heritage areas, defined by Murphy et al. [62],”.

Point 3: The authors inserted just three keywords. I think keywords could be extended to represent this manuscript's main contribution.

Response 3: We appreciate the Reviewer’s suggestion. The authors believe the reviewer's opinion is correct, so these terms have been expanded.

Point 4: Figures 8, 10, 12, 16, 18, and 20 should be improved. Additionally, figure 22 should be plotted larger and brighter than the present form.

Response 4: We appreciate the Reviewer’s suggestion. Figures have been improved. Most of the figures have been corrected.

Point 5: There is no need to show the interface environment of utilized software in figures 24 and 30. The authors could provide full-screen and clear images.

Response 5: We appreciate the Reviewer’s suggestion. The figures have been modified.

Point 6: The authors could follow the standard organization in preparing the introduction section. Please remove subsection (2. Related work) and design an integrated literature review without subsection. Besides, the authors are recommended to briefly mention the main contribution of the manuscript after previous works. In this present version, two paragraphs (lines 62 to 71 and 126 to 131) have been considered for this purpose.

Response 6: We appreciate the Reviewer’s suggestion. The authors have separated the introduction from the state of the art to establish a clear differentiation between the objectives set out in the paper and those investigations carried out up to now.

The authors have expanded the conclusions section in relation to the reviewer's comments. “In reference to all of the above and, taking into account the development of commercially available instruments, there are numerous investigations that focus their work on the use of small-dimensional scanners for the massive use of records of pieces from museums and archaeological sites. The value of this work lies in covering the knowledge gap about the applicability to the precision demand of low-cost 3D optical sensors. In addition, this document provides the evaluation of said 3D sensors under a BIM Platform environment through vector analysis, an analysis unpublished up to now.”  

Reviewer 3 Report (New Reviewer)

1. In this manuscript, two kinds of structured light scanners a high-cost Artec Spider 3D scanner, and the low-cost Revopoint POP 3D Scanner were compared with “Structure from Motion image-based data capture techniques”, however, the conclusion is not clear. So, it is suggested to further summarize.

2. The novelty and scientific significance of the manuscript were not well highlighted, such as algorithms, applications, or universal laws, etc. So, it is suggested supplementary explanation.

3. What is the reason for "Validate the accuracy of the equipment through a new case study" through the 42 year old woman sculpture?

4. Relevant concepts need to be clearly defined, such as HBIM, evaluation 1, Cloud Compare ICP algorithm, etc. Why are the estimated standard errors in Table 3 and Table 4 the same but different in Table 5?

5. "3. Discussion","3. Methodology”. The chapter number is repeated

6. The paper can be further polished to increase readability.

Author Response

Response to Reviewer 3 Comments

Point 1: In this manuscript, two kinds of structured light scanners a high-cost Artec Spider 3D scanner, and the low-cost Revopoint POP 3D Scanner were compared with “Structure from Motion image-based data capture techniques”, however, the conclusion is not clear. So, it is suggested to further summarize.

Response 1: We appreciate the Reviewer’s suggestion.

Point 2: The novelty and scientific significance of the manuscript were not well highlighted, such as algorithms, applications, or universal laws, etc. So, it is suggested supplementary explanation.

Response 2: We appreciate the Reviewer’s suggestion. The authors have expanded the section on the points marked in relation to the reviewer's comments.

Point 3: What is the reason for "Validate the accuracy of the equipment through a new case study" through the 42 year old woman sculpture?

Response 3: We appreciate the Reviewer’s suggestion. This document has been submitted to this journal previously with three other previous reviewers who gave their approval in the absence of validation in another case study. For this we have needed a time of two months to work on the new scan of the woman's bust and show the previous reviewer the validation data that, as it appears in the experiment, the values are very approximate.

Point 4: Relevant concepts need to be clearly defined, such as HBIM, evaluation 1, Cloud Compare ICP algorithm, etc. Why are the estimated standard errors in Table 3 and Table 4 the same but different in Table 5?

Response 4 a): We appreciate the Reviewer’s suggestion. The authors have expanded the section on the points marked in relation to the reviewer's comments.  “The BIM environment, in new construction and Historic Building Information Modelling (HBIM) in heritage areas, defined by Murphy et al. [62],”. “. The study focuses on four types of comparisons. i) Compare the degree of metric-dimensional precision through an Iterative Closest Point (ICP) algorithm, defined by Besl and Mckay in 1992,”.

Response 4 b): Regarding the estimated standard errors in Table 3 and Table 4 the same but different in Table 5, the authors. The evaluation level was determined when the researchers determined that residual point cleaning of the point sets had been exhausted in terms of the possibilities of further refining the results. This issue is clarified in the text”. In this case, Table 3 and Table 4 are done through an automatic procedure and also in Table 5, that is, the evaluation is carried out in such a way that in all three cases an automatic evaluation is taken (the study of the problem could have been completed here in the analysis). But at this point, the authors decided to check how the data set would work if we chose to do it manually, that is, using the approximation of both point clouds without any control points. The approximation is done in CloudCompare through the "cross section" and tabulating the "shift box" on the respective axes (x,y,z). This does not really work as a black box but is known in the realm of point-cloud approximation. And it is here, in Table 5, when it is believed appropriate to develop this methodology to give weight to the methodology.

Point 5: "3. Discussion","3. Methodology”. The chapter number is repeated

Response 5: We appreciate the Reviewer’s suggestion. Indeed, there is an error in the numbering that has already been corrected.

Point 6: The paper can be further polished to increase readability.

Response 6: We appreciate the Reviewer’s suggestion. The authors have expanded the conclusions section in relation to the reviewer's comments. “In reference to all of the above and, taking into account the development of commercially available instruments, there are numerous investigations that focus their work on the use of small-dimensional scanners for the massive use of records of pieces from museums and archaeological sites. The value of this work lies in covering the knowledge gap about the applicability to the precision demand of low-cost 3D optical sensors. In addition, this document provides the evaluation of said 3D sensors under a BIM Platform environment through vector analysis, an analysis unpublished up to now.”  

Round 2

Reviewer 1 Report (New Reviewer)

Thank you for taking my comments into consideration and addressing them. 

Author Response

We thank the anonymous Reviewer for her/his time spent to suggest improvements for our paper.

Reviewer 2 Report (New Reviewer)

Evaluation of geometric data registration of small objects from non-invasive techniques: Applicability to the HBIM field

Manuscript Number: sensors-2155536-v2

Comments:

1)      The authors have revised the manuscript according to the reviewers' comments. The overall structure and writing of this manuscript are acceptable. However, the authors are encouraged to enhance the readability of the context by implementing the following steps.

·         There are sequential short sentences in all sections, especially in the conclusions. For example, Thus, the two structured-light scanners enable acceptable precision” or “With a difference of 0.93 mm, operators and researchers in archeology and historical architecture could use low-cost structured-light scanning.” The authors are recommended to avoid short sentences. Besides, please summarize the conclusion section using clear and appropriate phrases.

·         There are still several grammatical and punctuation mistakes. I deeply recommend that the authors double-check the manuscript using Grammarly or another
language-checking software
.

2)      The authors should increase the clarity and resolution of the figures. For example, Figures 22, 24, 25, 30, and 33.

3)      I could not find any modifications in reorganizing the introduction section. Please recheck the latest comment that I submitted in the previous round.

Author Response

Response to Reviewer 2 Comments

Point 1:

The authors have revised the manuscript according to the reviewers' comments. The overall structure and writing of this manuscript are acceptable. However, the authors are encouraged to enhance the readability of the context by implementing the following steps.

There are sequential short sentences in all sections, especially in the conclusions. For example, “Thus, the two structured-light scanners enable acceptable precision” or “With a difference of 0.93 mm, operators and researchers in archeology and historical architecture could use low-cost structured-light scanning.” The authors are recommended to avoid short sentences. Besides, please summarize the conclusion section using clear and appropriate phrases.

There are still several grammatical and punctuation mistakes. I deeply recommend that the authors double-check the manuscript using Grammarly or another language-checking software.

 Response 1: We thank the anonymous Reviewer for her/his time spent to suggest improvements for our paper. The sentences have been edited: “Therefore, the difference of 0.93 mm between the two structured light scanners makes this variation in accuracy acceptable to operators and researchers in archeology and historical architecture, who could use low-cost structured light scanning”. The conclusions section has been modified and grammatical errors have been corrected.

Point 2: The authors should increase the clarity and resolution of the figures. For example, Figures 22, 24, 25, 30, and 33.

Response 2: We appreciate the Reviewer’s suggestion. Figures have been improved.

Point 3:   I could not find any modifications in reorganizing the introduction section. Please recheck the latest comment that I submitted in the previous round.

Response 3: We appreciate the Reviewer’s suggestion. The introduction has been rearranged.

This manuscript is a resubmission of an earlier submission. The following is a list of the peer review reports and author responses from that submission.

Round 1

Reviewer 1 Report

1) I tried to find their objectives, achievements, and what the authors did in the abstract; but unfortunately, I cannot find any related information.
2) I'm surprised to see abstract "the authors didn't write, what they did and why?"
3) The abstract requires major revisions to reflect the key ideas of this paper as well as the details of the proposed approach. Also, emphasis on the goals of this article.
4) Please in two-three points stress out what is the main contribution of this article.
5) The motivation should be detailed. The reviewer is unable to catch it.
6) Introduction needs to rewrite.
7) Authors are invited to add more recent and relevant references to better illustrate the novelty of this work.
8) The sentences were so declarative, so that I can find out the novelty contribution from the research.
9) There is an insufficient explanation about the research criteria about analyzing former research papers compared to this study.
10) Kindly refer to some recent literature

Author Response

Response to Reviewer 1 Comments

Point 1: I tried to find their objectives, achievements, and what the authors did in the abstract; but unfortunately, I cannot find any related information.

Response 1: We appreciate the Reviewer’s suggestion. The abstract and the introduction have been modified indicating the achievements made with the research.

Point 2: I'm surprised to see abstract "the authors didn't write, what they did and why?"

Response 2: We appreciate the Reviewer’s suggestion. The abstract has been modified in relation to what was requested by the reviewer.

Point 3: The abstract requires major revisions to reflect the key ideas of this paper as well as the details of the proposed approach. Also, emphasis on the goals of this article.

Response 3: We appreciate the Reviewer’s suggestion. The abstract has been modified. “Reverse engineering and creation of digital twins provides secure value to works of art for documenting, cataloging and maintenance control tracking in the field of cultural heritage. The replica in BIM models of the objects ensures the cultural interest every artistic work. 3D sensors called low-cost structured light scanners have evolved for multiple use in the entertainment market, but also for use as data acquisition and processing techniques for research purposes. Nowadays, and with the development of structured light data capture technologies, it is possible to capture the geometry of objects and acquire high-resolution 3D data sets at a very low cost. In this research, we work on an artistic model of small dimensions that have a certain singularity in their geometry, being representative of small objects and museum pieces. For this, the precision of two structured light scanners is analyzed, a high-cost equipment, such as the Artec Spider scanner, and another low-cost equipment, such as the Revopoint POP 3D Scanner, with the data acquisition technique based on short-range image capture such as photogrammetry. The accuracy of the data capture is evaluated through a mathematical algorithm and the segmentation of a set of points experimented with to verify the spatial resolution of the subsets of points. In addition, the precision of the geometry of the 3D model is analyzed through a vector analysis in a BIM environment, an unprecedented analysis until now. The work provides information on the precision of the equipment, through evaluation through algorithms and through the study of the density of points at a submillimeter scale. The results demonstrate the similarity of precision in this millimeter range, where photogrammetry obtained very positive results, reaching a precision of 0.70 millimeters with respect to the Artec Spider and 0.57 millimeters against the Revopoint POP 3D Scanner but in a morphometric analysis the results of the 3D geometry may vary depending on the records of the equipment”.

Point 4: Please in two-three points stress out what is the main contribution of this article.

Response 4: We appreciate the Reviewer’s suggestion. Highlights are provided in relation to what the reviewer requests.

“The precision of structured light equipment is analyzed with the photogrammetry technique.

The record was analyzed through a mathematical comparison algorithm and vector analysis in a BIM environment.

The results demonstrate the similarity of precision in this millimeter range of the three cases analysed.

In the study it was detected that the low-cost structured light scanner behaves very similar to the results obtained from photogrammetry.

With the applicability of the procedure used, different technologies can be compared, and the true geometry of heritage artifacts can be recorded”.

Point 5: The motivation should be detailed. The reviewer is unable to catch it.

Response 5: We appreciate the Reviewer’s suggestion. The following paragraph is added referring to the motivation. “3D scanning applied to archaeological objects or artifacts allows the shapes and their texture to be stored in digital format, speeding up the process of representation, identification and cataloging. On the other hand, with the new disruptive building information modeling technologies applied to archaeology, 3D artifacts pose a challenge for the area of knowledge of archeology and historical architecture, since these models can be inserted in the model as a parametric element”.

Point 6: Introduction needs to rewrite.

Response 6: We appreciate the Reviewer’s suggestion. The introduction has been modified. “There are several strategic lines dealing with the preservation, conservation, restoration and maintenance of the built environment of Cultural Heritage (CH). One of them is research, as it helps to assist in the knowledge of movable and immovable property. The development of techniques such as digital photogrammetry and the availability of structured light electronic equipment components play an essential role in the registration of objects and works of art. In line with the appearance of these new technologies, the complexity of the process in the conservation of cultural assets means that new perspectives of studies and implementations are constantly being formulated to achieve the recommendations of UNESCO; protect landscapes, natural environ-ments, and those created by man, which are of cultural or aesthetic interest, or which form a harmonious natural whole [1].  Choosing the right technology and equipment allows operators working with digital tools the correct workflow, as well as the quality of the results. Currently, low-cost 3D sensors are the most popular for entertainment purposes, but they are equally admissible for research work. Knowing how far the use of this equipment can go is a challenge in society and a research proposal in the area of archeology and architecture of small objects.

Thus, the scope of this research work is to compare the results obtained from an archaeological object record by means of Structure from Motion (SfM) and two equipment for capturing through structured light: high-cost equipment such as the Artec Spider scanner and other low-cost equipment such as the Revopoint POP 3D Scanner. In this context, this work develops an evaluation of the precision of low-cost equip-ment that currently appears on the market and high-quality equipment. To do this, we work on an artistic model of small dimensions that have a certain singularity in geometry, and that is representative of small objects and museum pieces. The applicability of these technologies that quickly capture and record data on the true geometry, serves to create predictive models of degradation, evaluation of painting in the framework of sculpture and, in general, detect pathologies in relation to the mineralogy of the stone. In addition, in this research, a digital twin of the object taken to a building information model (BIM) is processed from an evaluation of the resulting model. 3D scanning point cloud and photogrammetry data are used to create parametric BIM objects in complex shapes in the field of artistic sculpture”.

Point 7: Authors are invited to add more recent and relevant references to better illustrate the novelty of this work.

Response 7: We appreciate the Reviewer’s suggestion. More recent and relevant references have been added.

Point 8: The sentences were so declarative, so that I can find out the novelty contribution from the research.

Response 8: We appreciate the Reviewer’s suggestion. With the utmost respect to the reviewer, the authors don’t understand what he refers to when he speaks of "declarative sentences".

Point 9: There is an insufficient explanation about the research criteria about analyzing former research papers compared to this study.

Response 9: We appreciate the Reviewer’s suggestion. An explanation has been added about the research criteria:

“In the literature there are several studies in which archaeological objects are analyzed from their 3D model [19–22] determining in some cases, the similarity of geometric shapes between different objects. In the line of comparison of massive data acquisition techniques, Molero et al. [23] evaluated the use of structured light scanning photogrammetry through an Artec scanner. Kersten et al. [24] compared the geometric accuracy of portable 3D scanning systems, including the Artec Spider, with other equipment with similar features and prices. Other lines of work are related to the use of RGB-D cameras, for example, the Kinect v1 (Microsoft) released on the market around 2014. Lachat et al. [25] evaluated the accuracy of the characteristics of this sensor for 3D reconstruction of small objects. The use of the Artec Spider scanner and the Revopoint POP 3D Scanner has been used in the work of [26] to evaluate the accuracy of the size change of an implant of two skulls for additional surgical purposes, between a plastic model and the model human Morena et al. [27] worked on the precision of the low-cost EinScan-Pro in a sculpture by Eduardo Chillida. The creation of 3D models of small archaeological objects requires an effective methodology to capture the small geometric details [1]. Another of the big problems is the objects with cylindrical geometry where the capture of the registers must allow capturing the bot-tom of the objects as happens, for example, in this case. We must also add the number of new scanner products dedicated to capturing data from 3D representations or the characteristics of the capture camera and its calibration [5]. At this point and as far as it has been investigated, no work has been performed to date in which these data acquisition technologies and 3D models generated from an environment using the Building Information Modeling (BIM) methodology have been evaluated. And if low-cost 3D sensors have enough precision to be used reliably in the registration of museums and academic spaces that require the virtual reconstruction of 3D objects. Therefore, the work provides information on the precision of the equipment, through evaluation through algorithms and through the study of the density of points at a submillimeter scale.  This evaluation process is carried out in two different environments, one under a comparison algorithm used in scientific research such as Cloud Compare and a new framework through the BIM methodology, this workflow is described in the Figure 1”.

Point 10: Kindly refer to some recent literature

Response 10: We appreciate the Reviewer’s suggestion. Recent literature has been added. 6 new references have been included.

Reviewer 2 Report

To capture the geometry of objects and acquire high-resolution 3D data sets at a very low cost, authors have proposed an artistic model of small dimensions that have a certain singularity in their geometry, being representative of small objects and museum pieces. The accuracy of the data capture has been evaluated via a mathematical algorithm and the segmentation of a set of points was experimented with to verify the spatial resolution of the subsets of points. Besides, the precision of the geometry of the 3D model has been analyzed through vector analysis in a BIM environment.  The obtained results demonstrated that the similarity of precision is in millimeter range, whereas photo grammetry are positive, reaching a precision of 0.70 millimeters with respect to the Artec Spider and 0.57 millimeters against the Revopoint POP 29 3D Scanner.

            The results of the manuscript under consideration seem to be sound and correct, however, some of the minor issues with the manuscript are mentioned below:

1.      Abstract of the article is not concise and appealing as it does not summarize the study in a systematic fashion and even symbols have been used which is quite unusual.

2.      2. The motivation and recent literature are entirely missing in the introductory section as such authors are advised to revamp both of them to a great extent. In fact, it should be clarified why they consider this problem and what the advantages of the proposed method are.

3.      I would like to advocate for a more focused exposition that clearly and very visibly states the fundamental assumptions made and systematically makes clear where these assumptions enter.

4.      The reference list should be updated with a few more recent works closely related to the undertaken problem. 

Author Response

Response to Reviewer 2 Comments

To capture the geometry of objects and acquire high-resolution 3D data sets at a very low cost, authors have proposed an artistic model of small dimensions that have a certain singularity in their geometry, being representative of small objects and museum pieces. The accuracy of the data capture has been evaluated via a mathematical algorithm and the segmentation of a set of points was experimented with to verify the spatial resolution of the subsets of points. Besides, the precision of the geometry of the 3D model has been analyzed through vector analysis in a BIM environment.  The obtained results demonstrated that the similarity of precision is in millimeter range, whereas photo grammetry are positive, reaching a precision of 0.70 millimeters with respect to the Artec Spider and 0.57 millimeters against the Revopoint POP 29 3D Scanner.

The results of the manuscript under consideration seem to be sound and correct, however, some of the minor issues with the manuscript are mentioned below:

We appreciate the Reviewer’s suggestion. We thank the anonymous Reviewer for her/his time spent to suggest improvements for our paper.

Point 1: Abstract of the article is not concise and appealing as it does not summarize the study in a systematic fashion and even symbols have been used which is quite unusual.

Response 1: We appreciate the Reviewer’s suggestion. The abstract has been modified.

Point 2: The motivation and recent literature are entirely missing in the introductory section as such authors are advised to revamp both of them to a great extent. In fact, it should be clarified why they consider this problem and what the advantages of the proposed method are.

Response 2: We appreciate the Reviewer’s suggestion. “3D scanning applied to archaeological objects or artifacts allows the shapes and their texture to be stored in digital format, speeding up the process of representation, identification and cataloging. On the other hand, with the new disruptive building information modeling technologies applied to archaeology, 3D artifacts pose a challenge for the area of knowledge of archeology and historical architecture, since these models can be inserted in the model as a parametric element”.

Point 3: I would like to advocate for a more focused exposition that clearly and very visibly states the fundamental assumptions made and systematically makes clear where these assumptions enter.

Response 3: We appreciate the Reviewer’s suggestion. In relation to the issue raised by the reviewer, the paper has dealt with a systematic approach through the evaluation of a mathematical algorithm, the point density established by the point cloud and, furthermore, the concordance between the precision of the previous methodology and the study of vectors in the BIM environment: “Accuracy is evaluated through a mathematical algorithm and is experimented with by segmenting a set of points to verify the spatial resolution of the subsets. In addition, the precision of the geometry of the 3D model is analyzed through a profile analysis using vectors. The results determine the proximity of the data of the technical files and the knowledge of the variables that intervene in the different techniques used. The photo-grammetry obtained very positive results, reaching a precision of 0.70 millimeters with respect to the Artec Spider and 0.57 millimeters against the Revopoint POP 3D Scanner”.

Point 4: The reference list should be updated with a few more recent works closely related to the undertaken problem. 

Response 4: We appreciate the Reviewer’s suggestion. More recent and relevant references have been added.

Reviewer 3 Report

Please see the comments in the attachment.

Author Response

Response to Reviewer 3 Comments

This study presents an evaluation of three methods to create a point cloud of an archaeological object, namely a high-end laser scanner, low-cost laser scanner, and photogrammetric technique. The overall experiment is interesting and the description is in-detail. However, there are some comments that are needed for further clarification.

We appreciate the Reviewer’s suggestion. We thank the anonymous Reviewer for her/his time spent to suggest improvements for our paper.

Point 1: Will it be really fine to announce the names of the laser scanners in the experiment and compare them? It might be an attack for the company who owns an inferior one. Do the authors consult with the companies if they allow their products to be named? For example, in Line 626, it might cause a bad reputation for one of the laser scanner owned companies.

Response 1.: We appreciate the Reviewer’s suggestion. The authors consider that it is convenient to explicitly mention the brand of the equipment since otherwise the scientific community would not know the type to which we are referring. In all the cases that we have detected, the aggregation of the brand of the equipment has been observed, for example, (1). The specifications and data of the equipment have been verified to be correct and the experimentation reflects the reality of the results, regardless of whether it benefits or harms the characteristics of the equipment, therefore, there is no subjective bias of partiality that could imply the need to ask permission. The authors also think that it is not normal to request permission from the company to publish these results, since, if this were the case, not only in instruments but also in software, algorithms and methodologies, permission would have to be requested in all cases.

If the reviewer refers to this paragraph: “The results show that the Revopoint POP 3D Scanner obtains the highest number of points per surface unit, followed by photogrammetry and lastly the Artec Spider scanner”, the fact that a team obtains the highest number of points per surface unit, followed by photogrammetry and lastly the Artec Spider scanner Point density does not mean higher accuracy.

  • Kersten, T.P.; Lindstaedt, M.; Starosta, D. Comparative Geometrical Accuracy Investigations of Hand-Held 3d Scanning Systems - AN Update. ISPAr 2018, 422, 487–494, doi:10.5194/ISPRS-ARCHIVES-XLII-2-487-2018

Point 2: There is only one case study to be evaluated. Please kindly consider adding at least one more case to show the consistency of the experiment.

Response 2.: We appreciate the Reviewer’s suggestion. In relation to this opinion of the reviewer, the authors think that using another case study means doubling the length of the article, which is why it would not be publishable in most scientific journals and most of them have limits on its length. However, we consider the opinion of the reviewer and include it in future research.

Point 3: Figure 1, “ICP Algorithm” is wrongly spelled.

Response 3.: We appreciate the Reviewer’s suggestion. The word has been corrected.

Point 4: There is a need to readjust the manuscript because the lines are poorly displayed. For example, Line 124 should be on another page and numerous figures and tables are misplaced and appeared before mentioned in texts.

Response 4.: We appreciate the Reviewer’s suggestion. The manuscript has been readjusted.

 Point 5: Line 157 has an error maybe from copy-and-paste when preparing the manuscript.

Response 5.: We appreciate the Reviewer’s suggestion. The error has been corrected.

Point 6: Figure 4 has a red square in the figure. I know that it is because the authors click on one of the aligned photos in Metashape and should not raise any problems. Some readers may question what is the different red rectangle there so please consider editing this figure.

Response 6.: We appreciate the Reviewer’s suggestion. The figure 4 has been edited.

Point 7: Why is the quality in Table 2 set at high except at the build mesh step?

Response 7.: We appreciate the Reviewer’s suggestion. Regarding the processing parameter that is in “Build Mesh”, this step is not a necessary step that influences the results of the comparison, therefore, the decision to take quality media has been arbitrary.

Point 8: Why are there only two evaluations in Table 3 meanwhile other evaluations have at least 3 evaluations? It shows inconsistency in the experiment.

Response 8.: We appreciate the Reviewer’s suggestion. The evaluation level was determined when the researchers determined that residual point cleaning of the point sets had been exhausted in terms of the possibilities of further refining the results. This issue is clarified in the text.

Point 9: The caption of Table 5 is wrong.

Response 9.: We appreciate the Reviewer’s suggestion. The caption of Table 5 has been changed.

Point 10: Why the comparison of SfMv and SLSvp in Table 5 has 2 steps of evaluation? Will this affect the judgment for the quality of the laser scanners because this laser scanner has an extra step for evaluation? It might not reflect the real quality of the capturing by the low-cost scanner.

Response 10.: We appreciate the Reviewer’s suggestion. This procedure doesn’t affect the results between the comparison of SfMv and SLSvp shown in Table 5 due to what was previously described.

Point 11: What is the meaning of “Phase 2” in Line 354? Is it about the Best Fitting Manual step?

Response 11.: We appreciate the Reviewer’s suggestion. Yes, it is the best fit manual step.

Point 12: Line 385 and 450, the “Figure” here have a small f meanwhile, other Figure mentionings utilize capitalized F. Please consider the consistency of the format.

Response 12.: We appreciate the Reviewer’s suggestion. Changed the word "Figure" throughout the format.

Point 13: Line 388 has 2 “Distribution”.

Response 13.: We appreciate the Reviewer’s suggestion. The repeated word "distribution" has been removed.

Point 14: The caption of Figure 14 has too many “of” which makes it very confusing for the reader. Please consider rearranging it.

Response 14.: We appreciate the Reviewer’s suggestion. Modified text in Figure 14. “Distribution of subsets of representative points in the rim plane of the vessel”.

Point 15: Line 398 should add reference to SLSvs also. And mm2 needs to be superscript.

Response 15.: We appreciate the Reviewer’s suggestion. “The subsets of points SLVvs in the 400 mm2 area portion obtained a total of 455 points for, which is equivalent to a point density of 1.13 points/mm2. For the subset of SLSVP points, a total of 6752 points was obtained, which is equivalent to a density of 16.18 points/mm2. Finally, with the photogrammetry technique, a subset of SfMV points of 1465 points has been obtained, which is equivalent to a density of 3.66 points/mm2. Figure 15 shows the dispersion error of the subsets of points of the selected points. The points density was measured in square millimetres”. Superscript changed.

Point 16: There are a lot of self citations which will decrease the reliability of this experiment. Please consider changing the reference to other studies.

Response 16.: We appreciate the Reviewer’s suggestion. New studies has been added.

Point 17: Line 434, the authors mention testing with SLSvs and SfMv. Why not consider testing with SLSvp also?

Response 17.: We appreciate the Reviewer’s suggestion. The authors consider that it is not necessary to repeat the process with the remaining subsets of points, since what is intended is to demonstrate the methodology corresponding to the analysis of precision by vectors.

Point 18: Line 462, so the author also put SLSvp into the BIM environment. What is the difference between this step and the one in comment number 17?

Response 18.: We appreciate the Reviewer’s suggestion. The authors believe that the use of the SLSvp dataset is representative as a vector precision methodology.

Point 19: In Figure 18, why is there an outlier (1,983) in the data? Should not the test specimen be a pot or vase, in which point clouds should be equally distributed through the z-axis?

Response 19.: We appreciate the Reviewer’s suggestion. It is not an atypical value, but it is determining maximum and minimum values of deviation with respect to the section of vector planes.

Point 20: In Table 6, would not it be SLSvs, SLSvp, and SfMv for the list?

Response 20.: We appreciate the Reviewer’s suggestion. The text in table 6 is modified.

Point 21: In Line 505 “Surface analysis values of points in the subsets”is wordy and confusing. Please consider rephrasing it.

Response 21.: We appreciate the Reviewer’s suggestion. Has been reformulated by: “These analysis values”.

Point 22: In Line 518, Why does SLSvp points have an error? The authors should give more insights into this issue.

Response 22.: We appreciate the Reviewer’s suggestion. The error can be determined in the photograph itself where in the construction of the mesh the CloudCompare algorithm did not work, altering the morphology.

Point 23: In Line 598, there are 2 “stop”.

Response 23.: We appreciate the Reviewer’s suggestion. The repeated word "stop" has been removed.

Point 24: In Figure 25, why the Min dist. (mm) [black dots] the evaluation 2 and 3 are higher than Max dist. [red dots] and other data in the same evaluation?

Response 24.: We appreciate the Reviewer’s suggestion. Because the CloudCompare algorithm that establishes the nearest neighbor can determine positive points or negative points.

Point 25: In Line 681, I am afraid that you cannot infer POP laser scanner as a representative for other low-cost laser scanners. You must change this sentence.

Response 25.: We appreciate the Reviewer’s suggestion. This sentence has been changed by: “could Revopoint POP 3D Scanner be used as a low cost scanner”.

Round 2

Reviewer 1 Report

Accept as is.

Author Response

thank you very much, I appreciate your work as an expert reviewer

Author Response

Point 2:

Author’s response: It is a false claim saying that it will double the length of the article. The authors have to add only the experiment sample, result, and discussion parts. There is no need for other parts such as the introduction, literature review, methodology, and conclusion (if the results from your case studies are really consistent like what you presented from a lone present case study) to be modified. Hence, it is clearly not “doubling the article length”. Furthermore, adding at least one more case study will allow the readers to be ensured that the research result is consistent and reliable.

We appreciate the Reviewer’s suggestion. The reviewer has absolutely right with what he says. However, making a new case study means for the authors, in the first place, to request permission from the corresponding museum to be able to either lend the artifact or take all the equipment to the museum. This request for authorization is not immediate, you must have all the appropriate permissions to access the piece. It would take us a considerable amount of time at the administrative level. Secondly, it is necessary to establish in the introduction why these two pieces are chosen and what the selection criteria are. This would change the state of the art if an architectural piece is chosen that is less related to archeology is chosen. It would also be necessary to add a photograph with the control points on the ground to establish the couplings and procedures in the photogrammetry and the rest of the methodology. We would have to refer to the historical review and context of the piece, in addition to modifying the methodology and other points expressed by the reviewer. The authors estimate that the work may involve at least four months of work. The choice of the MDPI publisher has been made due to the speed of the evaluation, and, therefore, our objective would be modified "De facto" since it is intended that said article be published as background to a research project that deals with the capture and BIM modeling of archaeological objects and which will soon be presented in the R&D State Plan of the Spanish Ministry of Education. Finally, state that the changes made, which would be considerable, would modify the article and the rest of the reviewers might not agree. We humbly believe that the data extracted from the second case study would not be very significant from those expressed and obtained in this investigation. We ask the reviewer to take into account all these drawbacks.

Point 4:

Author’s response: There are still some parts that need to be modified. For example, Table 3 is still shown before mentioned.

We appreciate the Reviewer’s suggestion. It has been revised and modified.

Point 8:

Author’s response: The reviewer does not understand the response. Please answer more clearly to the comments.

We appreciate the Reviewer’s suggestion. Upon arriving at evaluation two, and by means of the point-cloud noise elimination procedure, the authors determined that better results were not achieved in the refinement process, so it was decided to finish the adjustment up to this level. 

Point 10:

Author’s response: The response you gave me did not explain anything more. How can comparisons between SPIDER vs SfM and POP vs SfM be compared if it does not have the same procedure? Why POP can have both automatic and manual alignment meanwhile, SPIDER can have only a type of alignment that also is not specified how it can be achieved? Moreover, what is the letter “y” that shows itself at the position that should be the word “and” in some of the captions?

We appreciate the Reviewer’s suggestion. Now, at this point, we have understood what the reviewer means. In the first review in section 8 it says: “Response 8.: We appreciate the Reviewer’s suggestion. The evaluation level was determined when the researchers determined that residual point cleaning of the point sets had been exhausted in terms of the possibilities of further refining the results. This issue is clarified in the text”.

In this case, Table 3 and Table 4 are done through an automatic procedure and also in Table 5, that is, the evaluation is carried out in such a way that in all three cases an automatic evaluation is taken (the study of the problem could have been completed here). analysis). But at this point, the authors decided to check how the data set would work if we chose to do it manually, that is, using the approximation of both point clouds without any control points. The approximation is done in CloudCompare through the "cross section" and tabulating the "shift box" on the respective axes (x,y,z). This does not really work as a black box but is known in the realm of point-cloud approximation. And it is here, in Table 5, when it is believed appropriate to develop this methodology to give weight to the methodology. As we understand that perhaps an explanation is necessary to give an explanation, the text referring to this step is modified

Point 14:

Author’s response: I fear that the text in the manuscript is not the same as what the authors provided here.

We appreciate the Reviewer’s suggestion. The caption of figure 14 has been modified.

Point 15:

Author’s response: 2 in mm2 is still not superscripted.

We appreciate the Reviewer’s suggestion. Superscripted has been changed.

Point 17&18:

Author’s response: So, what is the difference between “the methodology corresponding to the analysis of precision by vectors.” and “a vector precision methodology.”?

We appreciate the Reviewer’s suggestion. The authors apologize for expressing two terms that could be different. Answer 18 is not exactly “a vector precision methodology”, the authors refer to “analysis of precision by vectors”. Apologies.

Point 19:

Author’s response: “Outlier” in my meaning is that why the number is so high compared to other cells?

We appreciate the Reviewer’s suggestion. In relation, why is there an outlier (1,983) in the data?. This number is just as high compared to (0.992) without comparing it between the resulting average and the value (0.01) which is the minimum.

Point 22:

Author’s response: So, does it mean that the SLSvp point has lower quality than SLSvs because SLSvs does not have an error?

We appreciate the Reviewer’s suggestion. The reviewer will be able to verify in the photograph that the mesh is not flat, this is an obvious error in the CloudCompare algorithm and this does not mean that SLSvs has an error, since it can be clearly seen in Figure 22 that the surface is flat. It is about exposing the errors in the investigation when this happens.

Point 24:

Author’s response: Please let me ask again. Judging from Figure 24, the Max, Sigma, Avg., and Min align themselves as I can easily understand that they are sorted from the top (highest) to bottom (lowest). Then, what is the difference between the sequence of them in Figure 25? Why Min dist. is the highest one here at 0 mm.? So, if the Min dist. is 0 mm. and Max dist. is -0.4 mm., should not Min dist. becomes Max dist. and previous Max dist. becomes Min dist.? Could you please explain this figure?

We appreciate the Reviewer’s suggestion. It is evident that in Figure 24 there are no negative points

Point 25:

Author’s response: I am afraid that this sentence needs to be further rearranged. It does not make sense to me.

We appreciate the Reviewer’s suggestion. If the difference is 0.93mm, so many operators and researchers in the knowledge area of archeology and historical architecture could use this instrument (Revopoint POP 3D Scanner) at very low cost. 

Round 3

Reviewer 3 Report

This research requires additional experiments such as another case study to verify the consistency of the result. The authors try to convince that there is no time however, the authors can process all the document to ask for the permission to use other artifact since the first time they ask for only one of it. The authors stated that there would be difficulties about how to explain why they choose the new case study. Nevertheless, the reason why they choose this current case study is not explained plausibly. The responses from the authors also do not explain me more than when I comment to ask. Therefore, I would like to reject this manuscript.

Author Response

We appreciate the Reviewer’s suggestion. We thank the anonymous Reviewer for her/his time spent to suggest improvements for our paper.

The authors have carried out a new experimental campaign with another object to verify the consistency of the results. The study, the tables and figures, and the discussion of the results have been completely modified. A paper is attached in word format with the changes and a pdf where the modifications are highlighted.